# MULTI-MODAL DATA MIXTURES FOR VISION-LANGUAGE MODEL TRAINING

## ABSTRACT

Vision-Language models (VLMs) are typically trained on a diverse set of multi-modal domains, yet current practices rely on costly manual tuning. This paper introduces **MMix**, a principled framework for automatically determining multi-modal data mixtures for VLM training. We formulate this task as a modality-aware alignment maximization over domains, deriving multi-modal alignment scores from the dual solution through inter-modal coupling variables. Our method is crucially designed to handle domains with missing modalities, allowing for the systematic integration of language-only domains. In experiments on both 0.5B and 7B VLMs, **MMix** boosts accuracies on diverse evaluation benchmarks with marginal computational cost. Remarkably, it matches the expert-tuned performance $1.28\times$ faster in image-text tuning and extends to more complex multi-modal video scenarios outperforming uniform weights performance with only 33% steps.

## 1 INTRODUCTION

Vision-Language Models (VLMs) have advanced significantly with the availability of large-scale multi-modal datasets. The training data for VLMs is typically a complex mixture from numerous domains and multiple modalities (Bai et al., 2023b; Liu et al., 2023c; Li et al., 2024a; Liu et al., 2024a). For example, LLaVA-OneVision is trained on 20.6% Doc/Chart/Screen, 20.1% Math/Reasoning, and 8.9% OCR data, etc., and includes text and vision modalities (Li et al., 2024a). Since such domains help maintain and balance the skill distribution that a trained large multimodal model should cover (Li et al., 2024a), many studies follow the topic or capability-oriented rule with domain structure when collecting data, such as LLaVA (Liu et al., 2023c; Li et al., 2024a), Qwen (Bai et al., 2023a; Yang et al., 2025), LLAMA (Dubey et al., 2024), Gemini (Team et al., 2023), InstructBLIP (Dai et al., 2023), and others (Li et al., 2025; Tong et al., 2024; Chen et al., 2024e; Laurençon et al., 2024). Moreover, the composition of these domains critically impacts VLM effectiveness (Bai et al., 2023b; Li et al., 2024a; Liu et al., 2024b; Gadre et al., 2023). *"How to determine the proportions of each domain to improve VLMs' performance?"* is an essential question and remains an open challenge.

Existing strategies for constructing multimodal data mixtures often lack a formal methodology. Data recipes for many state-of-the-art models are not publicly released, while open-source models typically rely on expensive manual tuning or heuristic adjustments based on developers' experience (Bai et al., 2023b; Li et al., 2024a). For instance, Flamingo relies on empirically-tuned weights (Alayrac et al., 2022), LLaVA-NeXT manually adds data domains to improve specific skills (Liu et al., 2024b), and InstructBLIP uses a simple sampling heuristic (Dai et al., 2023) to handle data imbalance. Such approaches are inefficient, unscalable, and potentially suboptimal. Consequently, a principled and efficient methodology for optimizing the data mixture for VLMs is notably absent.

Although data mixing strategies have shown considerable success in Large Language Model (LLM) training (Xie et al., 2023; Fan et al., 2024b; Liu et al., 2024c; Kang et al., 2024), directly transferring these unimodal approaches to VLMs presents significant challenges due to their fundamental differences. The VLM data mixing problem introduces two unique challenges: *(i)* integrating features from **different modalities** (*e.g.,* text and vision); and *(ii)* handling domains with **missing modalities**, which frequently arises in VLM training where some domains include text-image paired data for visual learning, while others have text-only data for preserving linguistic abilities. Therefore, a specialized, modality-aware methodology is required for effective VLM data mixing.

In this paper, we introduce **MMix**, a framework for automatic weighting multimodal data domains for VLM training. **MMix** computes modality-aware alignment scores by formulating the multimodal data mixing problem as domain alignment maximization and deriving the scores in terms of the dual solution. We achieve cross-modal integration via shared latent variables that map multi-modal features into a common space. In addition, **MMix** handles missing modalities by ensuring they do not introduce noise in the alignment objective. The resulting scores directly translate into resampling weights, yielding improved generalization and higher efficiency without relying on costly manual tuning.

Specifically, the novelty and contribution of this work can be summarized as:

- We design the first automatic data mixing weighting strategy for VLMs. We introduce *modality-aware domain alignment scores* that serve as domain training weights. We formulate the data mixing problem as alignment maximization over domains with coupling multi-modal variables, where the alignment scores can be derived from the dual solution.

- Our method is designed to handle heterogeneous multi-modal data, which is a fundamental challenge for VLMs. It supports domains with differing modalities by ensuring incomplete data contributes no error to the alignment objective.

- We empirically validate our multi-modal data mixing method on 0.5B and 7B VLMs across diverse benchmarks, demonstrating its performance improvements and efficiency gains. Notably, it outperforms uniform weights with just 56% steps and achieves expert-tuned weights performance $1.28\times$ faster on the 0.5B model in image-text instruction tuning. It can scale to more complex settings including video modality, where it improves generalization over uniform weights with only 33% steps.

## 2 RELATED WORKS

**Data composition in VLMs.** The performance of modern VLMs is critically dependent on the composition of their training data. A standard practice in the field is to curate data into distinct, skill-oriented domains to ensure a balanced set of capabilities. For example, the development of the LLaVA family (Liu et al., 2023c; Li et al., 2024a; Liu et al., 2024a) involved explicitly adding new data domains like DocVQA and ChartQA to improve targeted skills such as OCR and chart understanding. They openly release the LLaVA-OneVision (Li et al., 2024a) datasets as collections of domain-specific data, which we use in our experiments. Similarly, the Qwen-VL (Bai et al., 2023b) and Gemini (Team et al., 2023) employ a multi-stage training pipeline that combines multi-modal data with text-only dialogue to maintain language capabilities. InstructBLIP (Gu et al., 2025) also groups 26 public datasets into 11 categories to cover a wide variety of tasks and capabilities. Many other works (Li et al., 2025; Tong et al., 2024; Chen et al., 2024e; Laurençon et al., 2024) follow such a capability-oriented rule to construct domains. While preliminary steps in the data pipeline such as data cleaning, toxicity removal, quality filtering, and coreset selection are also important aspects, our work focuses on the subsequent challenge of weighting the given pre-curated, skill-specific domains.

**Data mixing.** Despite the widespread practice of domain-structured data curation in VLMs, the subsequent step of determining the proportional mixture of these domains largely relies on developer intuition or costly empirical tuning. For instance, LLaVA-One (Li et al., 2024a) and Flamingo (Alayrac et al., 2022) manually tuned domain weights for their promising performance. Other approaches, like that for LLaVA-NeXT (Liu et al., 2024b), involve reactively adding new data to address perceived skill gaps, which is inefficient and heuristic. InstructBLIP (Gu et al., 2025) observes that ignoring the mixing problem in VLMs leads to unstable training and harms performance. While data mixing has been studied more formally for unimodal LLMs, these approaches are fundamentally ill-suited for VLMs. Most of them (Xie et al., 2023; Fan et al., 2024b; Liu et al., 2024c; Ye et al., 2024; Kang et al., 2024) rely on proxy models' training, which is difficult to combine in the multi-stage VLM pipeline. Recent directions (Xie et al., 2025; Zhang et al., 2025) integrate into LLM training, but they are not designed to handle multimodal features and cannot manage domains with missing modalities.

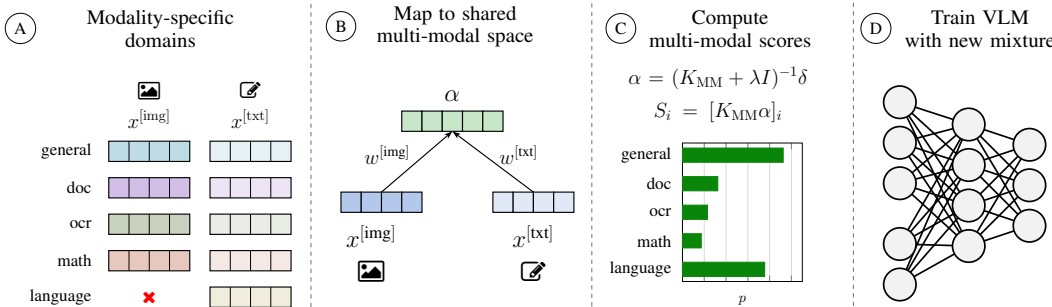

Figure 1: **Pipeline of multi-modal domain mixing for VLM training**. Modality-specific embeddings $x_i^{[v]}$ are extracted from the midstage trained model for each domain. Some domains may lack certain modalities (e.g., the language domain has no image data). The $k$ domains are then mapped to a shared multi-modal space by the latent variables $\alpha$ of the multi-modal alignment maximization problem (4). The multi-modal kernel matrix $K_{\mathrm{MM}}$ is computed as the pairwise inner products between domain embeddings across modalities via (5). Finally, (6) is applied to $K_{\mathrm{MM}}$ and $\alpha$ to obtain score $S_i$, $i = 1, \ldots, k$ indicating the multi-modal alignment of each domain. A resampling non-uniform distribution $p$ is obtained by softmax-normalizing the scores. Finally, image-text instruction tuning of the target VLM is carried out by sampling according to the obtained data mixture $p$.

## 3 MULTI-MODAL MIXING WITH MODALITY-AWARE DOMAIN ALIGNMENT

We propose **MMix**, an optimization-based framework for automatic multi-modal data mixing. VLM data presents unique challenges: feature heterogeneity and varying modality availability across domains. We address these by formulating mixing as an alignment maximization problem towards a shared signal direction. In Section 3.1, we derive the multi-modal alignment objective by coupling domain contributions in a *unified* latent space. In Section 3.2, we refine this formulation to handle missing modalities explicitly. The practical mixing pipeline of our method is shown in Figure 1.

**Setup and objective.** Let $\mathcal{D}_{\mathrm{MM}} = \{D_1, \ldots, D_k\}$ be the set of $k$ VLM training data domains (e.g., Math, OCR, etc.). These domains define the skill sets that the final trained VLM should possess. Data within a domain $D_i$ have the same modalities, while the modalities across domains could be different. Each sample $a^{[v]}$ from modality $v, v = 1, \ldots, V$ (e.g., vision or text) where $V$ is the number of modalities, can be represented through its semantic embedding $h^{(L)}(a^{[v]})$ extracted from the $L$-th hidden layer of the pretrained VLM $h$. The $i$-th domain embedding $x_i^{[v]} \in \mathbb{R}^d$, $v = 1, \ldots, V$, $i = 1, \ldots, k$ for the $v$-th modality can be constructed as the semantic centroid $x_i^{[v]} = \frac{1}{|D_i|} \sum_{a^{[v]} \in D_i} h^{(L)}(a^{[v]})$, which can effectively represent data domains thanks to the high-dimensional, non-linear representations learned by Transformers (Xie et al., 2025; Ling et al., 2025). The data mixing objective is to determine a domain weight vector $p \in \Delta_k$, where $\Delta_k$ is the probability simplex (Albalak et al., 2023; Fan et al., 2024b), enhancing the generalization performance of VLMs. The VLM is trained with the specific sampling probability $p_i$ for data from the $i$-th domain.

### 3.1 MULTI-MODAL DOMAIN ALIGNMENT SCORES

We first consider the case of mixing multiple domains containing only a single modality. Since there are multiple domains corresponding to various capabilities, the fundamental goal of VLM training is to equip the model with generalizable knowledge that can transfer across domains. Each domain $D_i$ is associated with an alignment score $S_i'$ with the projection direction $w$ within the embedding space optimally representing the general structure in all $k$ domains. By assigning a uniform target value of 1 for all domains, our optimization objective seeks a weight vector $w$ that exhibits strong alignment with the entire collection of domain embeddings $x_i$, $i = 1, \ldots, k$, rather than biasing it towards any specific one. This leads to the following primal optimization problem:

$$\min_{w,e} \frac{1}{2\lambda} \sum_{i=1}^{k} e_i^2 + \frac{1}{2} \|w\|_{\mathrm{F}}^2 \quad \text{s.t. } e_i = 1 - w^\top x_i, \ i = 1, \ldots, k, \tag{1}$$

where $w \in \mathbb{R}^d$ represents the projection vector, $e = [e_1, \ldots, e_k] \in \mathbb{R}^k$ denotes the individual projection errors for each domain, and $\lambda > 0$ is a regularization parameter.

**Interpretation.** The form (1) has a well-defined interpretation from the perspective of signal processing. It is analogous to a *beamformer* (Van Trees, 2002) where the mean embedding acts as the desired steering vector, representing the common structure shared across domains, and the covariance matrix represents the dispersion of the domains. Consequently, the optimal projection corresponds to a direction that balances maximizing alignment with the shared signal while minimizing interference through the covariance $(\Sigma + \lambda I_d)^{-1}$ operator. The resulting projection score $S_i' = w^\top x_i$ therefore quantifies how well each domain $x_i$ aligns with the robust, common-mode direction. A higher score indicates stronger alignment with the characteristics shared by the domains.

To enable the *multi-modal* integration, we first write a lower bound of the primal objective in (1) that yields equivalent solutions. Through introducing latent variables $\alpha_i'$ and the Fenchel-Young inequality $\frac{1}{2\lambda}e^2 + \frac{\lambda}{2}\alpha'^2 \geq e\alpha'$, $\forall e, \alpha' \in \mathbb{R}^k$ (Rockafellar, 1974; Suykens, 2017), we can express the primal problem of single modality as:

$$J = \frac{1}{2\lambda} \sum_{i=1}^{k} e_i^2 + \frac{1}{2} \|w\|_{\mathrm{F}}^2 \quad \text{s.t. } e_i = 1 - w^\top x_i, \ i = 1, \ldots, k$$

$$\geq \sum_{i=1}^{k} e_i \alpha_i' - \frac{\lambda}{2} \|\alpha'\|_{\mathrm{F}}^2 + \frac{1}{2} \|w\|_{\mathrm{F}}^2 \tag{2}$$

$$= \sum_{i=1}^{k} (1 - w^\top x_i)\alpha_i' - \frac{\lambda}{2} \|\alpha'\|_{\mathrm{F}}^2 + \frac{1}{2} \|w\|_{\mathrm{F}}^2 =: J_{\mathrm{SM}},$$

where $\alpha' = [\alpha_1', \ldots, \alpha_k'] \in \mathbb{R}^k$ is the vector of latent variables. By analyzing the stationary conditions of the lower-bound single-modality objective function $J_{\mathrm{SM}}$ with respect to $w$ and $\alpha'$ and eliminating the primal variable $w$, the following solution in the latent variables is obtained: $\alpha' = (K + \lambda I_k)^{-1} 1_k$, where $K \in \mathbb{R}^{k \times k}$ is the domain affinity kernel matrix with $K_{ij} = x_i^\top x_j$, $I_k$ is the $k \times k$ identity matrix, and $1_k$ is a $k \times 1$ column vector of ones. The unimodal domain score $S_i'$ can then be expressed as: $S_i' = [K(K + \lambda I)^{-1} 1_k]_i$, which is consistent with the result in terms of covariance obtained from the original problem (1) as in the derivation details in Appendix A.1 and Appendix A.2.

Importantly, such dual structure with explicit latent variables $\alpha_i'$ in Equation (2) facilitates the extension to **multi-modal integration**. Let $w^{[v]}$ be the projection weight for modality $v = 1, \ldots, V$. Define the alignment objective for each modality $v$ as $J_{\mathrm{SM}}^{[v]}(w^{[v]}, \alpha)$. We express the multi-modal scoring objective as

$$\tilde{J}_{\mathrm{MM}} = \sum_{v=1}^{V} J_{\mathrm{SM}}^{[v]}(w^{[v]}, \alpha) = \sum_{v=1}^{V} \sum_{i=1}^{k} (1 - (w^{[v]})^\top x_i^{[v]})\alpha_i - \frac{\lambda}{2} \sum_{v=1}^{V} \|\alpha\|_{\mathrm{F}}^2 + \frac{1}{2} \sum_{v=1}^{V} \left\| w^{[v]} \right\|_{\mathrm{F}}^2, \tag{3}$$

which implicitly sets $\alpha'^{[1]} = \cdots = \alpha'^{[V]} = \alpha$, giving the connections between the domain embeddings of each modality and the latent variables of a shared multi-modal latent space, realizing the inter-modality couplings.

**Interpretation.** The dual multi-modal objective (3) jointly optimizes the scores $(w^{[v]})^\top x_i^{[v]}$ for all domains and modalities. Specifically, $w^{[v]}$ learns to optimally align the domain embeddings within each modality. The first term of (3) can be interpreted as an energy function (Bengio et al., 2009) penalizing high-energy solutions, i.e., large $(1 - (w^{[v]})^\top x_i^{[v]})$ disagreements. The dual variable $\alpha_i$ serves as a consensus variable: large values push all modality weights to reduce disagreement for that domain. The remaining terms serve as regularization controlling the weight norm and the distribution of the dual variables.

### 3.2 MULTI-MODAL SCORES WITH MISSING MODALITIES

Accommodating data with incomplete modality coverage is a key challenge in VLM training. For instance, with vision and text modalities, some domains may only contain text, while others may

---

**Algorithm 1** Multi-modal Data Mixtures (**MMix**)

---

1: **Input:** Number of domains $k$, number of modalities $V$, domain embeddings $x_i^{[v]} \in \mathbb{R}^d$ for $i = 1, \ldots, k$ and available modalities $v = 1, \ldots, V$, and regularization parameter $\lambda$.

2: Fill the missing embeddings: set $x_i^{[v]} = 0_d$ for the unavailable modalities.

3: Construct kernel matrix: $K^{[v]} = [(x_i^{[v]})^\top x_j^{[v]}]_{i,j=1}^k$ for modality $v$.

4: Construct the multi-modal domain affinity matrix: $K_{\mathrm{MM}} = \sum_{v=1}^V K^{[v]}$.

5: Compute modality scores $S_i^{[v]} = \left[ K^{[v]}(K_{\mathrm{MM}} + \lambda I)^{-1} \delta \right]_i$.

6: Domain weights: $p_i = \frac{\exp(\sum_{v=1}^V S_i^{[v]})}{\sum_{j=1}^k \exp(\sum_{v=1}^V S_i^{[v]})}$.

7: **Output:** Domain weights $p = [p_1, \ldots, p_k]$.

---

present both. This scenario commonly occurs in practical settings as VLMs are typically trained on a mix of multi-modal and pure text data to retain the model's dialogue capabilities.

To address the issue of missing modalities, we adjust the projection errors appropriately. To be specific, we set $x_i^{[v]} = 0_d$ along with zero target for the missing modality $v$ in the $i$-th domain, which ensures that domains lacking a modality do not introduce spurious errors in the alignment objective. Therefore, the final multi-modal scoring objective from Equation (3) can be expressed as:

$$J_{\mathrm{MM}} = \sum_{v=1}^V \sum_{i=1}^k \left[ (\delta_i^{[v]} - (w^{[v]})^\top x_i^{[v]}) \alpha_i - \frac{\lambda}{2} \alpha_i^2 \right] + \frac{1}{2} \sum_{v=1}^V \left\| w^{[v]} \right\|_{\mathrm{F}}^2, \tag{4}$$

where $\delta_i^{[v]} \in \{0, 1\}$ indicates the existence of modality $v$ in domain $D_i$.

We obtain the solution in the shared latent variables $\alpha$ in the multi-modal setting by stationary conditions of (4) through the derivation in Appendix A.3, summarized in the following Proposition.

**Proposition 3.1** (Multi-modal Domain Alignment Scores). *Define the multi-modal kernel matrix as $K_{\mathrm{MM}} \in \mathbb{R}^{k \times k}$ with entries $K_{\mathrm{MM}_{ij}} = \sum_{v=1}^V K_{ij}^{[v]}$, with $K_{ij}^{[v]} = (x_i^{[v]})^\top x_j^{[v]}$. The optimal latent variables for the multi-modal alignment problem are given by:*

$$\alpha = (K_{\mathrm{MM}} + \lambda I)^{-1} \delta, \tag{5}$$

*where $\delta = [\delta_1, ..., \delta_k]^\top$ with entries $\delta_i = \sum_{v=1}^V \delta_i^{[v]}$. Note that $\delta_i$ is always a positive constant since all domains have at least one modality. At optimality, the domain alignment score $S_i^{[v]} = {w^{[v]}}^\top x_i^{[v]}$ for modality $v$ of domain $D_i$ in kernel representation is:*

$$S_i^{[v]} = \left[ K^{[v]}(K_{\mathrm{MM}} + \lambda I)^{-1} \delta \right]_i, \tag{6}$$

*with $K_{\mathrm{MM}}$ realizing the modality couplings.*

A high score $S_i^{[v]}$ indicates that the $v$-th modality of domain $D_i$ is well aligned with a common direction expressed through multi-modal coupling coefficients $\alpha_i$. After computing the scores $S_i^{[v]}$ for each modality $v$ of domain $D_i$, a single score is obtained for all the modalities by assembling the scores for each domain. The resampling distribution $p$ for VLM training is then obtained by softmax-normalizing the scores: $p_i = \frac{\exp(\sum_{v=1}^V S_i^{[v]})}{\sum_{i=1}^k \exp(\sum_{v=1}^V S_i^{[v]})}$.

**Computational complexity and practical implementation.** Our complete algorithm is summarized in Algorithm 1. Computing embeddings $x_i^{[v]}$ requires a cheap inference pass through the model from the previous stage. The kernel score computation (6) involves inverting a small $k \times k$ matrix, which is computationally cheap given the typically small number of domains $k$ used in VLM training. Notably, our method and operates independently of the VLM's optimization algorithm, enabling direct integration into existing training pipelines by simply adjusting sampling weights without modifying the underlying optimization procedure. This noninvasive approach is a key advantage in the VLM setting where many differing training pipelines are commonly used.

## 4 EXPERIMENTS

We conduct a comprehensive empirical evaluation of our multi-modal data mixing method for visual instruction tuning of LLaVA-OneVision (Li et al., 2024a) on diverse VLM benchmarks. We follow the standard domain construction of (Li et al., 2024a), with each domain corresponding to a target skill for a VLM. This domain-based structure is known to be crucial for balancing skill distribution, providing an ideal testbed for data mixing strategies (Laurençon et al., 2024; Dong et al., 2025). Furthermore, the data incorporate text, image, and video modalities and realistically reflects practical challenges where some modalities are absent in the domains.

First, we evaluate our method on the stage-2 image-text instruction tuning (Li et al., 2024a), which contains five domains including text and image modalities, and compare generalization on multiple benchmarks to other mixing baselines. **MMix** improves performance over expert-tuned mixtures at marginal computational cost. Further, we explore the transferability of our domain weights across model sizes. Then, we introduce an additional video modality in, showing that our algorithmic mixing naturally extends to more complex multi-modal settings, yielding consistent improvements and providing an efficient, scalable alternative to costly expert tuning.

**Training setup.** We train LLaVA-OneVision 0.5B and 7B models with batch size 128, sequence length 8192, and learning rate $10^{-5}$ with cosine decay. For experiments in Section 4.1, models are trained for 4500 steps following Li et al. (2024a) s.t. each example is used only once. The training data consists of five domains: General, Doc/Chart/Screen, Math/Reasoning, General OCR, and Language. The first four domains are structured as image-text pairs, while the Language domain consists of text data only, lacking the image modality. In Section 4.2, we introduce an additional VideoQA domain with video-text data and train for 3000 steps to further test our method's multi-modal capabilities.

**Baselines.** UNIFORM is the cost-free mixture assigning equal weights $p_i = \frac{1}{k}$, which, despite its simplicity, can be a strong baseline (Michel et al., 2021; Fan et al., 2024b). HUMAN corresponds to the domain weights manually optimized by the authors of (Li et al., 2024a). TEXT, IMAGE, and VIDEO represent weights derived solving Equation (1) based on embeddings from a single modality. If a domain lacks a specific modality, its corresponding weight is set to zero. AVG averages the domain weights of all single modalities. For example, AVG $= \frac{1}{2}$(TEXT + IMAGE) in Section 4.1 and AVG $= \frac{1}{3}$(TEXT + IMAGE + VIDEO) in Section 4.2. Moreover, FUSED are the domain weights computed from the fused multi-modal embedding, which is generated by the VLM after processing all modalities as a unified sequence. **MMix** computes the domain weights through Equation (6). The processes of embedding extraction and domain weight assignment are detailed in Appendix B.3.

**Evaluation benchmarks.** We use various benchmarks for evaluation of generalization in diverse tasks and they can be categorized into three classes following (Li et al., 2024a): *(1) Chart, Diagram, and Document Understanding.* Charts and diagrams are key formats for visual information expression. We evaluate the results on AI2D (Kembhavi et al., 2016), ChartQA (Masry et al., 2022), DocVQA (Mathew et al., 2021), and InfoVQA (Mathew et al., 2022), and OCRBench (Liu et al., 2024d) for text recognition. *(2) Perception and Multi-discipline Reasoning.* For more complex visual detection scenarios, we also evaluate on more challenging multi-disciplinary visual-language reasoning tasks. Specifically, we follow the multi-modal benchmarks of MME (Yin et al., 2023), MMBench (Liu et al., 2023d), and reasoning benchmarks including MathVerse (Zhang et al., 2024b), MMMU (Yue et al., 2024), and ScienceQA (Lu et al., 2022a). *(3) Real-world Understanding and Multi-modal Chatbots.* We also benchmark the capability of VLMs as a general-purpose assistant in the real world with specific benchmarks, including RealworldQA (x.ai, 2024) and MMStar (Chen et al., 2024c). In Table 4, we add two video benchmarks: Video-MMMU (Hu et al., 2025) and MVBench (Li et al., 2024c). We use the LLMs-Eval library (Zhang et al., 2024a) for evaluation.

### 4.1 **MMIX** IMPROVES PERFORMANCE ON BOTH 0.5B AND 7B VLMS

We train LLaVA-OneVision-0.5B using the domain reweighting strategies discussed in Baselines above during the single-image (i.e., no video) training phase. The domain weights are shown in Figure 2 (left) and listed in Appendix B.4. These models are evaluated on ten diverse benchmarks, with the 0-shot accuracy results presented in Table 1. Our **MMix** strategy achieves the highest average

score across all benchmarks, bringing a 1.24% improvement over UNIFORM. Importantly, it even surpasses HUMAN that requires large grid searches on the given domains with significant cost and is not scalable, while our method can find mixtures *automatically*. Remarkably, **MMix** learns faster: as shown in Figure 2 (right), it outperforms UNIFORM with just 56% steps and outperforms HUMAN with 78% steps, corresponding to $1.8\times$ and $1.28\times$ speedup factors, respectively.

For further analysis, as shown in Table 1, **MMix** outperforms 1) AVG that handles different modalities separately, and 2) FUSED that uses the fused embeddings from VLM with all available modalities as input. Moreover, **MMix** also surpasses unimodal strategies that ignore the information from other modalities, as demonstrated in Appendix B.5. This indicates the importance of distinctly considering the contributions of each modality and addressing the missing modal data specifically. Moreover, our ablation studies in Appendix B.6 demonstrate the robustness of **MMix**'s domain weights.

In addition, an interesting observation we find is that the downweighted domains do not result in sacrificing the model's corresponding capabilities. Specifically, even when **MMix** downweights Math and OCR domains compared with UNIFORM, it preserves the capabilities on MathVerse and OCRBench in Table 1. This suggests that our method supports positive transfer across domains, where *emphasizing a subset of high-alignment domains can promote emergent capabilities in others* as well, even when they receive less training weight.

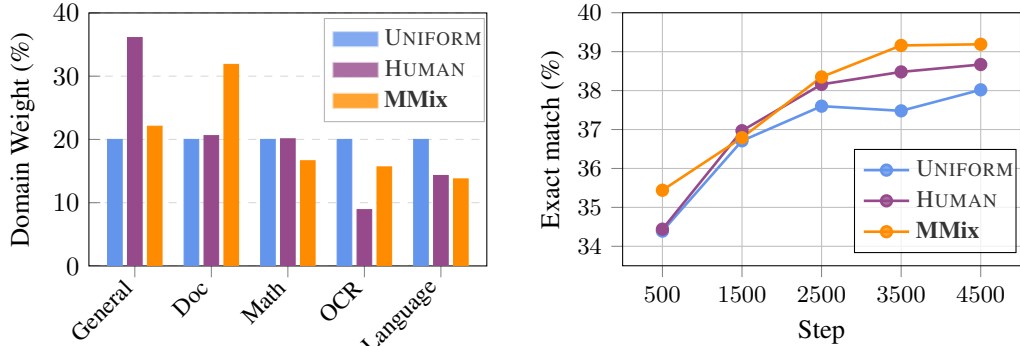

Figure 2: **Comparison of different data mixture strategies in the image-text instruction tuning.** (Left) Domain weights for UNIFORM, HUMAN, and **MMix**. (Right) Zero-shot average downstream accuracy of 0.5B models, where **MMix** achieves consistent improvement.

Table 1: **Comparison of data mixing strategies for LLaVA-0.5B image-text instruction tuning.** Results are reported as 0-shot accuracy across ten evaluation benchmarks. We compare our **MMix** against baselines: UNIFORM (equal weights), HUMAN (manual weights), AVG (averaged single-modality weights), and FUSED (weights from input concatenation). **MMix** improves performance on 8 out of 10 benchmarks over UNIFORM and 6 benchmarks over HUMAN.

| Benchmark | UNIFORM | HUMAN | AVG | FUSED | **MMix** |
|---|---|---|---|---|---|
| AI2D | $42.78_{\pm0.04}$ | $43.75_{\pm0.01}$ | $45.50_{\pm0.02}$ | $44.59_{\pm0.05}$ | $43.52_{\pm0.09}$ |
| DocVQA | $42.90_{\pm0.02}$ | $42.66_{\pm0.00}$ | $42.44_{\pm0.03}$ | $42.67_{\pm0.01}$ | $42.92_{\pm0.02}$ |
| InfoVQA | $22.25_{\pm0.03}$ | $22.61_{\pm0.07}$ | $22.43_{\pm0.04}$ | $23.50_{\pm0.03}$ | $22.13_{\pm0.05}$ |
| MathVerse | $18.27_{\pm0.03}$ | $17.26_{\pm0.11}$ | $18.32_{\pm0.06}$ | $19.29_{\pm0.08}$ | $18.91_{\pm0.07}$ |
| MMBench | $36.34_{\pm0.00}$ | $40.21_{\pm0.04}$ | $39.86_{\pm0.08}$ | $37.71_{\pm0.12}$ | $42.44_{\pm0.04}$ |
| MMStar | $33.45_{\pm0.06}$ | $36.04_{\pm0.10}$ | $33.50_{\pm0.14}$ | $34.44_{\pm0.20}$ | $35.88_{\pm0.03}$ |
| MMMU | $30.00_{\pm0.16}$ | $29.67_{\pm0.31}$ | $29.00_{\pm0.09}$ | $29.22_{\pm0.21}$ | $29.78_{\pm0.16}$ |
| ScienceQA | $62.42_{\pm0.02}$ | $65.84_{\pm0.02}$ | $64.80_{\pm0.04}$ | $63.46_{\pm0.09}$ | $64.50_{\pm0.01}$ |
| OCRBench | $45.30_{\pm0.05}$ | $44.60_{\pm0.09}$ | $45.30_{\pm0.06}$ | $43.50_{\pm0.09}$ | $45.80_{\pm0.05}$ |
| RealworldQA | $46.27_{\pm0.18}$ | $44.05_{\pm0.06}$ | $45.49_{\pm0.10}$ | $45.36_{\pm0.12}$ | $46.54_{\pm0.06}$ |
| Average | $38.00_{\pm0.09}$ | $38.67_{\pm0.12}$ | $38.66_{\pm0.08}$ | $38.37_{\pm0.12}$ | $\mathbf{39.24}_{\pm0.08}$ |
| Number over UNIFORM | - | 5/10 | 6/10 | 6/10 | 8/10 |

Table 2: **Transfer weights from LLaVA-0.5B to LLaVA-7B for image-text instruction tuning.** Results are reported as 0-shot accuracy across ten evaluation benchmarks. **MMix** improves performance on 8 out of 10 benchmarks over UNIFORM and 5 benchmarks over HUMAN.

| Benchmark | UNIFORM | HUMAN | AVG | FUSED | **MMix** |
|---|---|---|---|---|---|
| AI2D | $74.48_{\pm0.04}$ | $74.03_{\pm0.11}$ | $75.10_{\pm0.08}$ | $75.74_{\pm0.05}$ | $75.58_{\pm0.09}$ |
| DocVQA | $57.91_{\pm0.08}$ | $58.64_{\pm0.05}$ | $58.28_{\pm0.12}$ | $57.29_{\pm0.15}$ | $58.32_{\pm0.03}$ |
| InfoVQA | $34.76_{\pm0.15}$ | $35.91_{\pm0.09}$ | $36.95_{\pm0.07}$ | $36.06_{\pm0.11}$ | $36.23_{\pm0.18}$ |
| MathVerse | $29.31_{\pm0.09}$ | $26.85_{\pm0.14}$ | $27.33_{\pm0.18}$ | $28.68_{\pm0.06}$ | $28.55_{\pm0.12}$ |
| MMBench | $75.69_{\pm0.02}$ | $76.12_{\pm0.03}$ | $76.23_{\pm0.05}$ | $75.77_{\pm0.08}$ | $75.74_{\pm0.06}$ |
| MMStar | $49.04_{\pm0.11}$ | $50.26_{\pm0.16}$ | $50.44_{\pm0.09}$ | $49.46_{\pm0.14}$ | $50.19_{\pm0.10}$ |
| MMMU | $46.33_{\pm0.21}$ | $46.78_{\pm0.18}$ | $46.78_{\pm0.22}$ | $46.78_{\pm0.17}$ | $46.89_{\pm0.15}$ |
| ScienceQA | $87.31_{\pm0.06}$ | $90.38_{\pm0.02}$ | $89.53_{\pm0.04}$ | $85.52_{\pm0.09}$ | $90.23_{\pm0.07}$ |
| OCRBench | $56.80_{\pm0.13}$ | $57.30_{\pm0.08}$ | $56.70_{\pm0.11}$ | $56.60_{\pm0.08}$ | $57.90_{\pm0.14}$ |
| RealworldQA | $58.17_{\pm0.10}$ | $57.91_{\pm0.12}$ | $56.99_{\pm0.14}$ | $57.65_{\pm0.10}$ | $57.47_{\pm0.05}$ |
| Average | $56.98_{\pm0.11}$ | $57.42_{\pm0.11}$ | $57.43_{\pm0.13}$ | $56.96_{\pm0.11}$ | $\mathbf{57.71_{\pm0.11}}$ |
| Number over UNIFORM | - | 7/10 | 7/10 | 5/10 | 8/10 |

**Domain weights transfer to larger models.** Recent research on data mixing in text-only LLMs shows that domain weights derived from smaller models can be effectively transferred to larger ones (Xie et al., 2023; Fan et al., 2024b; Liu et al., 2024c). We investigate this phenomenon in VLMs. Specifically, we train 7B models with the domain weights obtained from 0.5B models. The evaluation results are presented in Table 2. Remarkably, **MMix** maintains its performance advantage over baselines even at this increased model scale, outperforming UNIFORM on 8 out of the 10 benchmarks.

**Marginal computational cost.** The computational overhead of our method is negligible, as we discussed *computational complexity* in Section 3. *(i)* Embedding extraction is a fast inference-only process. In our experiments, the embedding extraction takes 35 minutes on a single H100 GPU. *(ii)* Alignment score computation via (6) completes in seconds since the number of domains is small. The cost of our weight computation is marginal compared to the 90 and 620 GPU hours required to train 0.5B and 7B VLMs, respectively. Crucially, our approach also eliminates the need for expensive, time-consuming manual tuning of data mixtures, which is a key bottleneck in current VLM development.

Table 3: Computational cost is negligible relative to full model training. Cost in H100 GPU hours.

| Component | Cost (h) |
|---|---|
| Embedding extraction | 0.58 |
| Score computation | 0.01 |
| Total | 0.59 |
| Training (0.5B) | 90 |
| Training (7B) | 620 |

## 4.2 MMix SCALES TO MORE COMPLEX MULTI-MODAL SETTINGS

We further demonstrate the flexibility of **MMix** in more complex multimodal scenarios by adding a VideoQA domain that introduces video–text data. This creates a total of six domains with three modalities: text, image, and video. Our domain weights embeddings for this new configuration are shown in Figure 3 (left) and fully reported in Appendix B.7. We train 0.5B and 7B models with new domain weights and evaluate models on both image-only benchmarks (same pipeline as in Section 4.1) and benchmarks specifically designed to test video capabilities, namely MVBench (Li et al., 2024c) and Video-MMMU (Hu et al., 2025).

The results in Table 4 demonstrate that **MMix** achieves better performance over UNIFORM in this more complex setting as well. In addition, **MMix** achieves the best average performance, outperforming both AVG and FUSED. This verifies the effectiveness of our multi-modal construction compared to single-modality mixtures and simple early fusion. Notably, **MMix** achieves UNIFORM performance in only 33% steps, as shown in Figure 3 (right), resulting in a $3\times$ average speedup. Importantly, this experiment highlights the extensibility of our method to richer multi-modal configurations without

requiring manual efforts in costly grid searches; in fact, the expert-tuned HUMAN baseline was not available for this more complex setting, underscoring the practicality of *automatic* mixing.

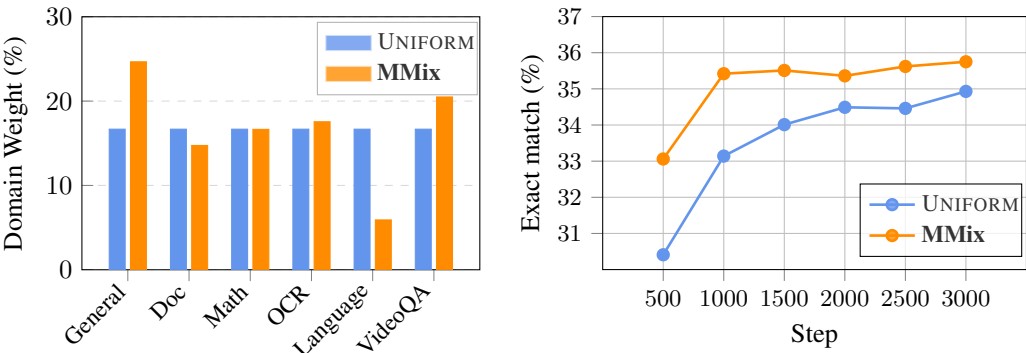

Figure 3: **Comparison of different data mixtures in the video-image-text instruction tuning.** (Left) Domain weights for UNIFORM and **MMix**. (Right) Zero-shot average downstream accuracy of 0.5B models, where **MMix** outperforms UNIFORM during the whole training process.

Table 4: **Comparison of data mixtures for LLaVA-0.5B/7B video-image-text instruction tuning.** Results are reported as 0-shot accuracy across twelve evaluation benchmarks. **MMix** achieves the best average performance on two model sizes. Full results with standard deviations are in Appendix B.8.

| Benchmark | 0.5B | | | | 7B | | | |
| --- | --- | --- | --- | --- | --- | --- | --- | --- |
| | UNIF. | AVG | FUSED | **MMix** | UNIF. | AVG | FUSED | **MMix** |
| AI2D | 41.68 | 42.81 | 42.84 | 42.88 | 71.83 | 72.41 | 72.83 | 72.15 |
| DocVQA | 42.20 | 41.68 | 41.29 | 42.54 | 56.47 | 56.42 | 55.67 | 57.51 |
| InfoVQA | 21.65 | 21.97 | 21.17 | 22.40 | 35.74 | 34.65 | 34.40 | 35.89 |
| MathVerse | 15.61 | 15.62 | 17.77 | 15.10 | 25.63 | 25.52 | 24.75 | 26.40 |
| MMBench | 34.36 | 26.80 | 35.14 | 34.45 | 71.05 | 75.52 | 73.28 | 74.57 |
| MMStar | 30.43 | 35.54 | 36.14 | 33.97 | 48.18 | 49.03 | 46.55 | 48.79 |
| MMMU | 30.00 | 29.78 | 30.44 | 29.78 | 45.67 | 45.11 | 44.78 | 45.56 |
| ScienceQA | 60.29 | 60.29 | 59.40 | 61.03 | 83.44 | 86.07 | 83.29 | 87.26 |
| OCRBench | 45.30 | 43.20 | 46.60 | 45.00 | 56.50 | 56.90 | 57.60 | 57.20 |
| RealworldQA | 47.19 | 46.41 | 46.27 | 47.32 | 57.91 | 56.99 | 59.22 | 57.39 |
| Video-MMMU | 13.78 | 13.78 | 12.78 | 13.84 | 29.78 | 30.56 | 29.11 | 30.33 |
| MVBench | 36.67 | 36.50 | 37.02 | 40.70 | 52.73 | 51.58 | 53.12 | 53.60 |
| Average | 34.93 | 34.53 | 34.74 | **35.75** | 52.91 | 53.39 | 52.88 | **54.40** |
| # over UNIF. | - | 4/12 | 7/12 | 9/12 | - | 6/12 | 5/12 | 10/12 |

## 5 CONCLUSION

This paper presents a principled approach to the key problem of automatically optimizing sampling weights across pre-defined domains for vision-language model training with negligible additional computational cost. Our formulation through modality-aware alignment maximization with coupling inter-modal variables addresses fundamental challenges in VLM training: handling missing modalities, optimizing cross-modal alignment, and determining domain mixing weights without costly grid searches. Empirical evaluations demonstrate that our method outperforms on average both uniform and manually-tuned mixtures across diverse VLM benchmarks with marginal computational cost. Our approach allows direct integration with existing diverse VLM training pipelines and making it valuable for practical applications. By automating domain reweighting, our method offers a path towards more data- and compute-efficient VLM training.

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

# A  PROBLEM FORMULATION

## A.1  FORMULATION FOR SINGLE MODALITY SETTING

Let the data mixture problem consist of $k$ data domains and their domain embeddings $x_i \in \mathbb{R}^d$ with their target $y_i \in \mathbb{R}$, $i = 1, \dots, k$. We first write the primal domain alignment problem for single modality:

$$\min_{w,e} \frac{1}{2\lambda} \sum_{i=1}^{k} e_i^2 + \frac{1}{2} \|w\|^2 \quad \text{s.t. } e_i = y_i - w^\top x_i, \, i = 1, \dots, k, \tag{7}$$

where $w \in \mathbb{R}^d$, $e = [e_1, \dots, e_k] \in \mathbb{R}^k$ are the projections, $\lambda > 0$ is a regularization constant.

From the Lagrangian with dual variables $\nu$:

$$\mathcal{L}(w, e; \nu) = \frac{1}{2\lambda} \sum_{i=1}^{k} e_i^2 + \frac{1}{2} \|w\|^2 - \sum_{i=1}^{k} \nu_i (e_i - y_i + w^\top x_i),$$

one takes the conditions for optimality, which are given as

$$\begin{cases} \dfrac{\partial \mathcal{L}}{\partial w} = w - \sum_{i=1}^{k} \nu_i x_i = 0 \quad \Longrightarrow \quad w = \sum_{i=1}^{k} \nu_i x_i, \\[2mm] \dfrac{\partial \mathcal{L}}{\partial e_i} = \dfrac{1}{\lambda} e_i - \nu_i = 0 \quad \Longrightarrow \quad e_i = \lambda \nu_i, \quad \forall i \\[2mm] \dfrac{\partial \mathcal{L}}{\partial \nu_i} = e_i - y_i + w^\top x_i = 0 \quad \Longrightarrow \quad \lambda \nu_i - y_i + \sum_{i=j}^{k} \nu_j x_j^\top x_i = 0 \quad \forall i. \end{cases}$$

Eliminating $w$ in the last condition gives the dual solution:

$$K\nu = y - \lambda\nu,$$
$$(K + \lambda I)\nu = y,$$
$$\nu = (K + \lambda I)^{-1} y,$$

where we defined the kernel matrix as $K = [x_i^\top x_j]_{i,j=1}^{k}$, and the target vector $y = [y_1, \dots, y_k]^\top$.

We are now ready to define the alignment score of domain $i$ as $S_i' = w^\top x_i$ in its kernel form:

$$S_i' = w^\top x_i = \left( \sum_{j=1}^{k} \nu_j x_j \right)^\top x_i = \left[ K(K + \lambda I)^{-1} y \right]_i. \tag{8}$$

### A.1.1  PRIMAL AND DUAL SCORE REPRESENTATIONS

We can write Equation (7) in the unconstrained form:

$$\min_{w} \frac{1}{2\lambda} \sum_{i=1}^{k} (y_i - w^\top x_i)^2 + \frac{1}{2} \|w\|^2.$$

This is a ridge regression problem where the target vector is $y = [y_1, \dots, y_k]^\top$. Let $X$ be the $k \times d$ data matrix with rows $x_1^\top, x_2^\top, \dots, x_k^\top$. Then the objective becomes:

$$\min_{w} \frac{1}{2\lambda} \|y - Xw\|^2 + \frac{1}{2} \|w\|^2.$$

The solution to this ridge regression problem is:

$$w = (X^\top X + \lambda I)^{-1} X^\top y.$$

The alignment score for domain $i$ is $S_i' = w^\top x_i$. The vector of alignment scores can be computed as $S' = Xw$. Substituting the expression for $w$:

$$S_i' = [X(X^\top X + \lambda I)^{-1} X^\top y]_i.$$

This is equivalent to (8) by standard matrix identity, i.e., Woodbury identity. The following remark summarizes the computational aspect of the primal and dual representations of the alignment score.

*Remark* A.1 (Efficient computation of the alignment score). The primal solution is written in terms of the covariance $X^\top X$, while the dual solution is in terms of the kernel matrix $XX^\top$. In the context of data mixture with large VLMs, the embedding dimension $d$ may be very large, so it is computationally advantageous to work in the dual with complexity $\mathcal{O}(k^3)$ where the number of data domains $k$ is typically much smaller. We report the computational time of the alignment score on a single A100 GPU in Table 5 as supporting evidence.

Table 5: Computational cost of alignment score is negligible. Time on a single A100 GPU.

| Number of Domains | Time (s) |
|---|---|
| 10 | 0.07 |
| 100 | 0.09 |
| 1000 | 0.48 |
| 10000 | 21.85 |

### A.2 INTRODUCING LATENT VARIABLES

We first give a lower bound to the objective (7) and introduce latent variables $\alpha_i'$, which will be used to couple the domains in the multi-modal setting. Starting from the primal single-modal problem (7), the following lower bound holds:

$$J = \frac{1}{2\lambda} \sum_{i=1}^{k} e_i^2 + \frac{1}{2} \|w\|_{\mathrm{F}}^2 \quad \text{s.t. } e_i = y_i - w^\top x_i, \ i = 1, \ldots, k$$

$$\geq \sum_{i=1}^{k} e_i \alpha_i' - \frac{\lambda}{2} \|\alpha'\|_{\mathrm{F}}^2 + \frac{1}{2} \|w\|_{\mathrm{F}}^2 \tag{9}$$

$$= \sum_{i=1}^{k} (y_i - w^\top x_i)\alpha_i' - \frac{\lambda}{2} \|\alpha'\|_{\mathrm{F}}^2 + \frac{1}{2} \|w\|_{\mathrm{F}}^2 =: J_{\mathrm{SM}},$$

where $\lambda > 0$ is a regularization constants and $J_{\mathrm{SM}}$ is the single modality objective. The above bound is based on the property that for two arbitrary vectors $e, \alpha'$ one has $\frac{1}{2\lambda}e^2 + \frac{\lambda}{2}\alpha'^2 \geq e\alpha'$, $\forall e, \alpha' \in \mathbb{R}^k$. The inequality can be verified using the Schur complement by writing in its quadratic form:

$$\frac{1}{2} \begin{bmatrix} e^T & \alpha'^\top \end{bmatrix} \begin{bmatrix} \frac{1}{\lambda}I & I \\ I & \lambda I \end{bmatrix} \begin{bmatrix} e \\ \alpha' \end{bmatrix} \geq 0.$$

From the Schur complement, it states the condition $\frac{1}{2}(\lambda I - I(\lambda I)I) \geq 0$, which proves the above inequality. This is also known as conjugate feature duality (Suykens, 2017) or the Fenchel–Young inequality for quadratic functions (Rockafellar, 1974).

Through the inequality, we have introduced latent variables, i.e. $\alpha_i'$, into the objective. We proceed by studying the stationary condition of $J_{\mathrm{SM}}$.

$$\begin{cases} \dfrac{\partial J_{\mathrm{SM}}}{\partial w} = -\sum_{i=1}^{k} \alpha_i' x_i + w = 0 \quad \Rightarrow \quad w = \sum_{i=1}^{k} \alpha_i' x_i, \\[2mm] \dfrac{\partial J_{\mathrm{SM}}}{\partial \alpha_i'} = y_i - w^\top x_i - \lambda \alpha_i' = 0 \quad \Rightarrow \quad \alpha_i' = \frac{1}{\lambda}\left(y_i - w^\top x_i\right) \quad \forall i. \end{cases} \tag{10}$$

By eliminating $w$ in (10), we obtain

$$w^\top x_i = \left(\sum_{j=1}^{k} \alpha_j' x_j\right)^\top x_i = \sum_{j=1}^{k} \alpha_j' \left(x_j^\top x_i\right) \quad \forall i.$$

Thus the solution in the latent variables is

$$\alpha_i' = \frac{1}{\lambda}\left(y_i - \sum_{j=1}^{k} \alpha_j' \left(x_j^\top x_i\right)\right)$$

$$\alpha' = (K + \lambda I)^{-1} y.$$

The score of domain $i$, i.e., $S_i = w^\top x_i$, writes in terms of the latent variables as:

$$S_i' = w^\top x_i = \left( \sum_{j=1}^k \alpha_j' x_j \right)^\top x_i = \sum_{j=1}^k \alpha_j'(x_j^\top x_i) = [K(K + \lambda I)^{-1} y]_i, \tag{11}$$

which matches (8) obtained by the original problem (7). Let $y = 1_k$, it recovers the uni-modal alignment score in Section 3.1.

### A.3 PROOF OF PROPOSITION 3.1

We first characterize the stationary points of $J_{\text{MM}}$ defined in Equation (4), as the stationary conditions lead to the optimal solution in the dual of the multi-modal problem. Note that the coupling across modalities can be achieved by creating a common latent space (Houthuys et al., 2018; Tao et al., 2024), i.e., by introducing the same latent variables $\alpha$ across all modalities in $J_{\text{MM}}$. By taking the partial derivatives of the weights $w^{[v]}$ and the latent variables $\alpha$, the conditions of the stationary points leading to MM scores are characterized by:

$$\begin{cases} \dfrac{\partial J_{\text{MM}}}{\partial w^{[v]}} = -\sum_{i=1}^k \alpha_i x_i^{[v]} + w^{[v]} = 0 \implies w^{[v]} = \sum_{i=1}^k \alpha_i x_i^{[v]} \\[2mm] \dfrac{\partial J_{\text{MM}}}{\partial \alpha_i} = \sum_{v=1}^V \left( \delta_i^{[v]} - (w^{[v]})^\top x_i^{[v]} \right) - \lambda \alpha_i = 0 \\[2mm] \Rightarrow \sum_{v=1}^V \delta_i^{[v]} - \sum_{v=1}^V \left( \sum_{j=1}^k \alpha_j \underbrace{(x_j^{[v]})^\top x_i^{[v]}}_{K_{ij}^{[v]}} \right) - \lambda \alpha_i = 0 \\[2mm] \Rightarrow \sum_{v=1}^V \delta_i^{[v]} - \sum_{j=1}^k \alpha_j \sum_{v=1}^V K_{ij}^{[v]} - \lambda \alpha_i = 0 \\[2mm] \Rightarrow \sum_{v=1}^V \delta_i^{[v]} - \sum_{j=1}^k \alpha_j K_{\text{MM}_{ij}} - \lambda \alpha_i = 0, \text{ where } K_{\text{MM}_{ij}}^{[v]} = \sum_{v=1}^V K_{ij}^{[v]}. \end{cases} \tag{12}$$

Define the multi-modal kernel matrix as $K_{\text{MM}} \in \mathbb{R}^{k \times k}$ with entries $K_{\text{MM}_{ij}} = \sum_{v=1}^V K_{ij}^{[v]}$, with $K_{ij}^{[v]} = (x_i^{[v]})^\top x_j^{[v]}$. The above conditions can be rewritten in matrix form as:

$$(K_{\text{MM}} + \lambda I)\alpha = \delta,$$

where $\delta \in \mathbb{R}^k$ is the vector with entries $\delta_i = \sum_{v=1}^V \delta_i^{[v]}$ with $\delta_i^{[v]} \in \{0, 1\}$ representing the existence of the modality $v$ of the domain $i$. The solution in the latent variable therefore is

$$\alpha = (K_{\text{MM}} + \lambda I)^{-1}\delta.$$

We can compute the domain alignment score for each modality as $S_i^{[v]} = {w^{[v]}}^\top x_i^{[v]}$. For modality $v$, at optimality:

$$w^{[v]} = \sum_{j=1}^k \alpha_j x_j^{[v]} \quad \Rightarrow \quad S_i^{[v]} = {w^{[v]}}^\top x_i^{[v]} = \sum_{j=1}^k \alpha_j (x_j^{[v]})^\top x_i^{[v]}.$$

Substituting $\alpha = (K_{\text{MM}} + \lambda I)^{-1}\delta.$, it yields in matrix form:

$$S_i^{[v]} = \left[ K^{[v]}(K_{\text{MM}} + \lambda I)^{-1}\delta \right]_i,$$

which is the multi-modal alignment score of domain $i$ for modality $v$. The ensemble score of domain $i$ then considers all modalities as $S_i = \sum_{v=1}^V S_i^{[v]}$.

### A.4 EIGENDECOMPOSITION PERSPECTIVE OF ALIGNMENT

Alignment score functions as a robust steering direction, suppressing domain-specific noise. In fact, we can assert the following:

**Lemma A.2.** *Let $K_{MM} \in \mathbb{R}^{k \times k}$ be the multi-modal kernel matrix with SVD $K_{MM} = U \Sigma U^\top$, where $U = [u_1, \ldots, u_k]$ are the singular vectors and $\sigma_1 \geq \cdots \geq \sigma_k$ are the singular values. The alignment score $S_i$ derived from the MMix objective (Proposition 3.1) is given by:* $S_i = \sum_{j=1}^{k} \left( \frac{\sigma_j}{\sigma_j + \lambda} \right) (u_j^\top \delta)(u_j)_i$.

Therefore, our score applies a spectral soft thresholding filter to the domain distribution. The alignment operator dampens the noisy directions (small eigenvalues) and effectively measures the projection of domain $i$ onto the robust semantic subspace (corresponding to large eigenvalues).

## B ADDITIONAL EXPERIMENTS

### B.1 DIFFERENCE BETWEEN IMAGE AND TEXT MODALITIES

In the multi-modality training process, there are two main challenges from the data perspective: *(i)* domains may have different modalities, and *(ii)* each domain's data features may vary significantly as captured by different modalities.

We take the LLaVA-OneVision dataset (LLaVA-OneVision-Data, 2024) as an example. The LLaVA-OneVision dataset includes five domains along with two modalities, image and text. Four domains include both image and text modalities, while the "Language" domain only has text. We visualize the embedding similarity matrix for text and image modalities independently in Figure 4. It shows that the domain relationships in different modalities can vary considerably, although the kernel magnitudes remain comparable. Note that the kernel matrices can be normalized to unit trace (Bach et al., 2004) if scales differed significantly.

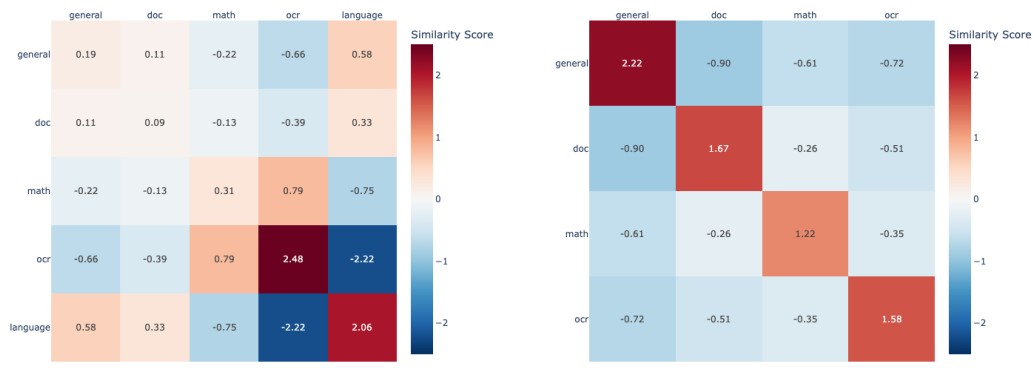

(a) Text modality.                    (b) Image modality.

Figure 4: Embedding kernel similarity matrix for different modalities.

### B.2 EXPERIMENTAL SETUP

We use the LLaVA-OneVision publicly available data (LLaVA-OneVision-Data, 2024) for training and follow the domain segmentation in the LLaVA-OneVision paper (Li et al., 2024a). We use 3 seeds for the benchmark evaluations and report the standard deviations.

Note that some training datasets used in (Li et al., 2024a) were not released and some datasets use different naming conventions than (Li et al., 2024a). Our specific domain settings are:

- **General:** aokvqa (cauldron,llava_format) (Schwenk et al., 2022), clevr (cauldron,llava_format) (Johnson et al., 2017), hateful_memes (cauldron,llava_format) (Kiela et al., 2020), image_textualization (filtered) (Pi et al., 2024), iconqa (cauldron,llava_format) (Lu et al., 2021b), IconQA (MathV360K) (Lu et al., 2021b), scienceqa (cauldron,llava_format) (Saikh et al., 2022), scienceqa (nona_context) (Saikh et al., 2022), st_vqa (cauldron,llava_format) (Xia et al., 2023), tallyqa (cauldron,llava_format) (Acharya et al., 2019), VisualWebInstruct (filtered) (Jia et al., 2025), visual7w (cauldron,llava_format) (Zhu et al., 2016), vistext (cauldron) (Tang et al., 2023), VizWiz (MathV360K) (Gurari et al., 2018), vqarad (cauldron,llava_format) (Lau et al., 2018), vsr (cauldron,llava_format) (Liu et al., 2023a), websight (cauldron) (Laurençon et al., 2024), allava_instruct_laion4v (Chen et al., 2024a), allava_instruct_vflan4v (Chen et al., 2024a), vision_flan (filtered) (Xu et al., 2024b), intergps (cauldron,llava_format) (Lu et al., 2021a), llavar_gpt4_20k (Zhang et al., 2023b), sharegpt4o (Chen et al., 2024b), sharegpt4v (coco) (Chen et al., 2024b), sharegpt4v (knowledge) (Chen et al., 2024b), sharegpt4v (llava) (Chen et al., 2024b), sharegpt4v (sam) (Chen et al., 2024b)
- **Doc/Chart/Screen:** ai2d (cauldron,llava_format) (Kembhavi et al., 2016), ai2d (gpt4v) (Kembhavi et al., 2016), ai2d (internvl) (Kembhavi et al., 2016), chart2text (cauldron) (Kantharaj et al., 2022), chartqa (cauldron,llava_format) (Masry et al., 2022), diagram_image_to_text (cauldron), dvqa (cauldron,llava_format) (Kafle et al., 2018), figureqa (cauldron,llava_format) (Kahou et al., 2017), hitab (cauldron,llava_format) (Cheng et al., 2021), infographic_vqa (Mathew et al., 2022), infographic_vqa_llava_format (Mathew et al., 2022), screen2words (cauldron) (Wang et al., 2021), tqa (cauldron,llava_format) (Kembhavi et al., 2017), ureader_cap (Ye et al., 2023), ureader_ie (Ye et al., 2023), robut_sqa (cauldron) (Ghosh et al., 2024), robut_wikisql (cauldron) (Ghosh et al., 2024), robut_wtq (cauldron,llava_format) (Ghosh et al., 2024), visualmrc (cauldron) (Tanaka et al., 2021), infographic (gpt4v) (Mathew et al., 2022), lrv_chart (Liu et al., 2023b), mapqa (cauldron,llava_format) (Chang et al., 2022), multihiertt (cauldron) (Zhao et al., 2022)
- **Math/Reasoning:** CLEVR-Math (MathV360K) (Lindström & Abraham, 2022), FigureQA (MathV360K) (Kahou et al., 2017), GEOS (MathV360K) (Seo et al., 2015), GeoQA+ (MathV360K) (Anand et al., 2024), Geometry3K (MathV360K) (Lu et al., 2021a), MapQA (MathV360K) (Chang et al., 2022), Super-CLEVR (MathV360K) (Li et al., 2023), TabMWP (MathV360K) (Lu et al., 2022b), UniGeo (MathV360K) (Chen et al., 2022), geo170k (align) (Gao et al., 2023), geo170k (qa) (Gao et al., 2023), geomverse (cauldron) (Kazemi et al., 2023), mavis_math_metagen (Zhang et al., 2024c), mavis_math_rule_geo (Zhang et al., 2024c), lrv_normal (filtered) (Liu et al., 2023b), geo3k (Lu et al., 2021a), raven (cauldron) (Zhang et al., 2019), PMC-VQA (MathV360K) (Zhang et al., 2023a), tabmwp (cauldron) (Lu et al., 2022b)
- **General OCR:** chrome_writting (Wendler & Gambot, 2023), hme100k (Yuan et al., 2022), iam (cauldron) (Marti & Bunke, 2002), iiit5k (Mishra et al., 2012), k12_printing, rendered_text (cauldron) (Wendler & Gambot, 2023), textcaps (Sidorov et al., 2020), textocr (gpt4v) (Singh et al., 2021), sroie, orand_car_a
- **Language:** magpie_pro (l3_80b_mt), magpie_pro (l3_80b_st), magpie_pro (qwen2_72b_st) (Xu et al., 2024a)
- **Video:** academic_qa, youtube (Zhang et al., 2024d), ActivityNetQA (Yu et al., 2019), NeXT-QA Xiao et al. (2021), PerceptionTest (Pătrăucean et al., 2023)

### B.3 Embedding extraction and domain weight assignment

For embedding computation, we use the pretrained LLaVA-OneVision model that has completed stage-1.5 pre-training and we randomly sample a subset of data from each domain. Given the presence of multiple datasets per domain, we extracted embeddings for 512 samples from each individual dataset. These sample embeddings were then averaged to create a single representation for each dataset. Subsequently, we averaged these dataset-level embeddings to capture the overall character of its respective domain. Then, we use domain-level embeddings to compute domain weights.

Once we compute the domain weights $p_i$ using Algorithm 1, our training sampling strategy takes dataset size into account as follows. We sample datasets proportionally to their size within each domain, and then sample individual data points uniformly from the chosen dataset. This results in the final sampling probability for a dataset $DS$ in domain $D_i$ being $P = \frac{|DS|}{|D_i|}p_i$, followed by uniformly sampling over instances in $DS$.

## B.4 DOMAIN WEIGHTS FOR THE IMAGE-TEXT INSTRUCTION TUNING (SECTION 4.1)

We report domain weights for Section 4.1 with five domains and two modalities in Table 6. Note that AVG = $\frac{1}{2}$ (TEXT+IMAGE). IMAGE$^\dagger$ sets its Language weight as same as HUMAN and reweight the others in IMAGE.

Table 6: **VLM Mixtures for the image-text instruction tuning.** Domain weights of different mixing strategies. IMAGE$^\dagger$ sets its Language weight as same as HUMAN and reweight the others in IMAGE.

| Domain | UNI. | HUMAN | TEXT | IMAGE | AVG | FUSED | **MMix** | IMAGE$^\dagger$ |
|---|---|---|---|---|---|---|---|---|
| General | 20.00 | 36.10 | 20.90 | 35.66 | 28.28 | 14.74 | 22.09 | 30.56 |
| Doc/Chart/Screen | 20.00 | 20.60 | 43.28 | 29.49 | 36.29 | 40.95 | 31.86 | 25.27 |
| Math/Reasoning | 20.00 | 20.10 | 15.24 | 17.92 | 16.58 | 20.21 | 16.63 | 15.36 |
| General OCR | 20.00 | 8.90 | 10.22 | 16.93 | 13.58 | 14.14 | 15.66 | 14.51 |
| Language | 20.00 | 14.30 | 10.35 | 0.00 | 5.18 | 9.95 | 13.76 | 14.30 |

## B.5 PERFORMANCE OF DOMAIN WEIGHTS COMPUTED BY SINGLE MODALITY

We add two unimodal strategies, TEXT and IMAGE$^\dagger$, in Tables 7 and 8 as addition for Tables 1 and 2. These two unimodal methods compute the domain weights derived from single modality in Section 3.1, based solely on text or image embeddings. Note that the Language domain does not have image data, thus IMAGE has 0% on this domain. For a more reasonable comparison, we set its Language domain weight to the same as HUMAN and reweight the others, finalizing to IMAGE$^\dagger$ in Table 6. Importantly, **MMix** still outperforms these unimodal strategies.

Table 7: **Comparison of data mixing strategies for LLaVA-0.5B image-text instruction tuning.** Results are reported as 0-shot accuracy across ten evaluation benchmarks. **MMix** achieves the best average performance, including the single-modality methods.

| Benchmark | UNIFORM | HUMAN | AVG | FUSED | **MMix** | TEXT | IMAGE$^\dagger$ |
|---|---|---|---|---|---|---|---|
| AI2D | 42.78 | 43.75 | 45.50 | 44.59 | 43.52 | 45.95 | 44.33 |
| DocVQA | 42.90 | 42.66 | 42.44 | 42.67 | 42.92 | 43.08 | 42.42 |
| InfoVQA | 22.25 | 22.61 | 22.43 | 23.50 | 22.13 | 23.45 | 21.47 |
| MathVerse | 18.27 | 17.26 | 18.32 | 19.29 | 18.91 | 16.50 | 18.53 |
| MMBench | 36.34 | 40.21 | 39.86 | 37.71 | 42.44 | 35.82 | 39.00 |
| MMStar | 33.45 | 36.04 | 33.50 | 34.44 | 35.88 | 34.19 | 34.67 |
| MMMU | 30.00 | 29.67 | 29.00 | 29.22 | 29.78 | 27.89 | 30.67 |
| ScienceQA | 62.42 | 65.84 | 64.80 | 63.46 | 64.50 | 64.60 | 63.86 |
| OCRBench | 45.30 | 44.60 | 45.30 | 43.50 | 45.80 | 45.30 | 45.20 |
| RealworldQA | 46.27 | 44.05 | 45.49 | 45.36 | 46.54 | 45.36 | 46.67 |
| Average | 38.00 | 38.67 | 38.66 | 38.37 | **39.24** | 38.21 | 38.69 |
| # over UNIFORM | - | 5/10 | 6/10 | 6/10 | 8/10 | 5/10 | 7/10 |

## B.6 ABLATION STUDIES

**Regularization parameter $\lambda$.** The parameter $\lambda$ is related to the degree of regularization. Despite this control, Table 9 demonstrates that our obtained domain weights are largely stable with respect to changes in $\lambda$.

**Number of samples for embedding extraction.** As we discussed in Appendix B.3, we sample a subset of datasets for embedding extraction. We test the robustness of domain weights with respect to the number of samples. The domain weights based on 256, 512, or 1024 samples from each individual dataset are reported in Table 9, which confirms that the domain weights obtained are stable regardless of the number of samples.

Table 8: **Comparison of data mixing strategies for LLaVA-7B image-text instruction tuning.** Results are reported as 0-shot accuracy across ten evaluation benchmarks. **MMix** achieves the best average performance, including single-modality methods.

| Benchmark | UNIFORM | HUMAN | AVG | FUSED | **MMix** | TEXT | IMAGE[†] |
|---|---|---|---|---|---|---|---|
| AI2D | 74.48 | 74.03 | 75.10 | 75.74 | 75.58 | 75.42 | 75.58 |
| DocVQA | 57.91 | 58.64 | 58.28 | 57.29 | 58.32 | 58.73 | 57.86 |
| InfoVQA | 34.76 | 35.91 | 36.95 | 36.06 | 36.23 | 36.83 | 36.22 |
| MathVerse | 29.31 | 26.85 | 27.33 | 28.68 | 28.55 | 26.14 | 27.83 |
| MMBench | 75.69 | 76.12 | 76.23 | 75.77 | 75.74 | 75.60 | 76.98 |
| MMStar | 49.04 | 50.26 | 50.44 | 49.46 | 50.19 | 49.51 | 50.72 |
| MMMU | 46.33 | 46.78 | 46.78 | 46.78 | 46.89 | 47.11 | 45.67 |
| ScienceQA | 87.31 | 90.38 | 89.53 | 85.52 | 90.23 | 86.91 | 90.08 |
| OCRBench | 56.80 | 57.30 | 56.70 | 56.60 | 57.90 | 57.70 | 57.50 |
| RealworldQA | 58.17 | 57.91 | 56.99 | 57.65 | 57.47 | 57.65 | 57.39 |
| Average | 56.98 | 57.42 | 57.43 | 56.96 | **57.71** | 57.16 | 57.59 |
| # over UNIFORM | - | 7/10 | 7/10 | 5/10 | 8/10 | 6/10 | 6/10 |

**Embedding aggregation.** Except for averaging the dataset-level averaged embeddings to represent each domain, another way is to aggregate dataset-level embeddings to domain embeddings according to their dataset sizes. Basically, sum the dataset-level embeddings reweighted by their sizes as domain weights. The domain weights computed by these two strategies are highly similar, as reported in Table 9.

**Model sizes.** We use the domain weights obtained from the LLaVA-0.5B model's embeddings and test the transferability for LLaVA-7B shown in Table 2. We compute domain weights based on the embeddings from LLaVA-7B as well. The result reported in Table 10 demonstrates that domain weights are stable across model sizes.

**The number of training steps for the pretrained model.** Regarding the sufficiency of feature extraction, the latent representations are from the mid-stage checkpoint of LLaVA-OneVision (Li et al., 2024b), which is a well-trained model. We additionally conducted a stability analysis by continuing training this checkpoint on its public mid-training data for 500 and 1000 additional steps and recomputed domain weights using our method. The results, presented in Table 10, demonstrate that the domain weights remain stable throughout training.

**Number of domains.** We run additional experiments with a reduced number of domains. We exclude 'General' from the original five domains, and the new domain weights obtained by **MMix** are: 26.7% Doc/Chart/Screen, 28.7% Math/Reasoning, 31.6% General OCR, and 13.0% Language. The domain weights of UNIFORM are 25% per domain. Table 11 demonstrates that **MMix** consistently shows a higher average accuracy. This validates the robustness of our method across different numbers of domains. Furthermore, the experiment in Section 4.2, which introduces a Video domain, demonstrates that **MMix** remains effective as the number of domains changes.

Table 9: **Domain weights across $\lambda$ values, and number of samples.** We observe that our method is robust to the choice of $\lambda$, the number of samples used, and two embedding aggregation methods.

| Domain | $\lambda$ Values | | | Number of Samples | | | Aggregate embeddings | |
|---|---|---|---|---|---|---|---|---|
| | 1 | 10 | 100 | 256 | 512 | 1024 | Equally | Dataset sizes |
| General | 21.57 | 22.09 | 23.14 | 24.48 | 22.09 | 23.96 | 22.09 | 23.20 |
| Doc/Chart/Screen | 28.90 | 31.86 | 34.79 | 27.32 | 31.86 | 30.84 | 31.86 | 30.98 |
| Math/Reasoning | 17.47 | 16.63 | 15.95 | 18.71 | 16.63 | 17.32 | 16.63 | 17.80 |
| General OCR | 16.74 | 15.66 | 14.49 | 16.72 | 15.66 | 17.04 | 15.66 | 16.91 |
| Language | 15.32 | 13.76 | 11.64 | 12.77 | 13.76 | 10.84 | 13.76 | 11.11 |

Table 10: **Domain weights across model sizes and the number of steps for pretrained checkpoints.** We observe that our method is robust to both model sizes and the number of steps.

| Domain | Model sizes | | # training steps for pretrained model | | |
|---|---|---|---|---|---|
| | 0.5B | 7B | Pretrained model | +500 steps | +1000 steps |
| General | 22.09 | 21.15 | 22.09 | 21.65 | 22.59 |
| Doc/Chart/Screen | 31.86 | 29.91 | 31.86 | 30.12 | 29.96 |
| Math/Reasoning | 16.63 | 19.64 | 16.63 | 18.74 | 18.82 |
| General OCR | 15.66 | 17.06 | 15.66 | 15.17 | 15.82 |
| Language | 13.76 | 12.23 | 13.76 | 14.33 | 12.99 |

Table 11: **Comparison of data mixtures on 4 domains for LLaVA-0.5B image-text instruction tuning. MMix** is robust across different numbers of domains.

| Benchmark | UNIFORM | MMix |
|---|---|---|
| AI2D | 42.75 | 43.75 |
| DocVQA | 40.79 | 41.71 |
| InfoVQA | 22.89 | 22.96 |
| MathVerse | 17.51 | 18.27 |
| MMBench | 30.76 | 33.59 |
| MMStar | 35.68 | 33.24 |
| MMMU | 28.89 | 31.78 |
| ScienceQA | 53.99 | 54.09 |
| OCRBench | 43.30 | 44.70 |
| RealworldQA | 38.30 | 42.88 |
| Average | 35.48 | **36.69** |
| Number over UNIFORM | - | 9/10 |

### B.7    DOMAIN WEIGHTS FOR VIDEO-IMAGE-TEXT INSTRUCTION TUNING (SECTION 4.2)

We report domain weights for Section 4.2 with six domains and three modalities in Table 12. Note that AVG = $\frac{1}{3}$ (TEXT+IMAGE+VIDEO).

Table 12: **VLM Mixtures.** Domain weights across different mixing strategies for three modalities.

| Domain | UNIFORM | TEXT | IMAGE | VIDEO | AVG | FUSED | **MMix** |
|---|---|---|---|---|---|---|---|
| General | 16.67 | 16.62 | 35.66 | 0.00 | 17.43 | 10.77 | 24.66 |
| Doc/Chart/Screen | 16.67 | 14.42 | 29.49 | 0.00 | 16.70 | 13.20 | 14.74 |
| Math/Reasoning | 16.67 | 30.73 | 17.92 | 0.00 | 16.22 | 9.44 | 16.65 |
| General OCR | 16.67 | 9.27 | 16.93 | 0.00 | 8.73 | 38.60 | 17.55 |
| Language | 16.67 | 13.89 | 0.00 | 0.00 | 4.63 | 16.73 | 5.91 |
| Video | 16.67 | 15.08 | 0.00 | 100.00 | 38.36 | 11.26 | 20.49 |

### B.8    TABLE 4 WITH STANDARD DEVIATIONS

We show the results with standard deviations of Table 4 in Tables 13 and 14.

### B.9    DOMAIN WEIGHTS TRANSFER TO QWEN2-VL

To explore the generality of **MMix** across different architectures, we test domain weights obtained from LLaVA-0.5B on Qwen-VL-2B (Wang et al., 2024) with the same setup of Section 4.2. The results show that **MMix** maintains its benefit compared with other baselines also on Qwen-VL-2B. It would be interesting to further explore the benefit of new domain weights on more model architectures.

Table 13: **Comparison of data mixtures for LLaVA-0.5B video-image-text instruction tuning.**

| Benchmark | UNIFORM | AVG | FUSED | **MMix** |
|---|---|---|---|---|
| AI2D | $41.68_{\pm0.08}$ | $42.81_{\pm0.09}$ | $42.84_{\pm0.10}$ | $42.88_{\pm0.04}$ |
| DocVQA | $42.20_{\pm0.06}$ | $41.68_{\pm0.05}$ | $41.29_{\pm0.06}$ | $42.54_{\pm0.08}$ |
| InfoVQA | $21.65_{\pm0.07}$ | $21.97_{\pm0.06}$ | $21.17_{\pm0.08}$ | $22.40_{\pm0.10}$ |
| MathVerse | $15.61_{\pm0.10}$ | $15.62_{\pm0.14}$ | $17.77_{\pm0.11}$ | $15.10_{\pm0.08}$ |
| MMBench | $34.36_{\pm0.02}$ | $26.80_{\pm0.03}$ | $35.14_{\pm0.06}$ | $34.45_{\pm0.04}$ |
| MMStar | $30.43_{\pm0.05}$ | $35.54_{\pm0.08}$ | $36.14_{\pm0.06}$ | $33.97_{\pm0.04}$ |
| MMMU | $30.00_{\pm0.15}$ | $29.78_{\pm0.09}$ | $30.44_{\pm0.13}$ | $29.78_{\pm0.11}$ |
| ScienceQA | $60.29_{\pm0.11}$ | $60.29_{\pm0.10}$ | $59.40_{\pm0.12}$ | $61.03_{\pm0.09}$ |
| OCRBench | $45.30_{\pm0.12}$ | $43.20_{\pm0.07}$ | $46.60_{\pm0.08}$ | $45.00_{\pm0.15}$ |
| RealworldQA | $47.19_{\pm0.18}$ | $46.41_{\pm0.16}$ | $46.27_{\pm0.12}$ | $47.32_{\pm0.10}$ |
| Video-MMMU | $13.78_{\pm0.08}$ | $13.78_{\pm0.04}$ | $12.78_{\pm0.10}$ | $13.84_{\pm0.06}$ |
| MVBench | $36.67_{\pm0.06}$ | $36.50_{\pm0.10}$ | $37.02_{\pm0.10}$ | $40.70_{\pm0.12}$ |
| Average | $34.93_{\pm0.10}$ | $34.53_{\pm0.09}$ | $34.74_{\pm0.10}$ | $\mathbf{35.75}_{\pm0.09}$ |
| Number over UNIFORM | - | 4/12 | 7/12 | 9/12 |

Table 14: **Comparison of data mixtures for LLaVA-7B video-image-text instruction tuning.**

| Benchmark | UNIFORM | AVG | FUSED | **MMix** |
|---|---|---|---|---|
| AI2D | $71.83_{\pm0.03}$ | $72.41_{\pm0.08}$ | $72.83_{\pm0.06}$ | $72.15_{\pm0.06}$ |
| DocVQA | $56.47_{\pm0.04}$ | $56.42_{\pm0.06}$ | $55.67_{\pm0.08}$ | $57.51_{\pm0.10}$ |
| InfoVQA | $35.74_{\pm0.12}$ | $34.65_{\pm0.10}$ | $34.40_{\pm0.07}$ | $35.89_{\pm0.06}$ |
| MathVerse | $25.63_{\pm0.11}$ | $25.52_{\pm0.12}$ | $24.75_{\pm0.08}$ | $26.40_{\pm0.14}$ |
| MMBench | $71.05_{\pm0.03}$ | $75.52_{\pm0.06}$ | $73.28_{\pm0.04}$ | $74.57_{\pm0.05}$ |
| MMStar | $48.18_{\pm0.08}$ | $49.03_{\pm0.06}$ | $46.55_{\pm0.04}$ | $48.79_{\pm0.07}$ |
| MMMU | $45.67_{\pm0.14}$ | $45.11_{\pm0.10}$ | $44.78_{\pm0.12}$ | $45.56_{\pm0.13}$ |
| ScienceQA | $83.44_{\pm0.04}$ | $86.07_{\pm0.13}$ | $83.29_{\pm0.10}$ | $87.26_{\pm0.08}$ |
| OCRBench | $56.50_{\pm0.11}$ | $56.90_{\pm0.08}$ | $57.60_{\pm0.09}$ | $57.20_{\pm0.14}$ |
| RealworldQA | $57.91_{\pm0.16}$ | $56.99_{\pm0.15}$ | $59.22_{\pm0.08}$ | $57.39_{\pm0.09}$ |
| Video-MMMU | $29.78_{\pm0.07}$ | $30.56_{\pm0.05}$ | $29.11_{\pm0.08}$ | $30.33_{\pm0.06}$ |
| MVBench | $52.73_{\pm0.08}$ | $51.58_{\pm0.12}$ | $53.12_{\pm0.07}$ | $53.60_{\pm0.11}$ |
| Average | $52.91_{\pm0.09}$ | $53.39_{\pm0.10}$ | $52.88_{\pm0.08}$ | $\mathbf{54.40}_{\pm0.10}$ |
| Number over UNIFORM | - | 6/12 | 5/12 | 10/12 |

### B.10 ALIGNMENT SCORE VS ORTHOGONAL SCORE

To further ablate the alignment benefit introduced in Section 3, we introduce the "Orthogonal Score" to quantify the uniqueness of different domains, i.e., downweighting high-alignment domains. Specifically for the domain $j$, set $e_i = \gamma_i - w^\top x_i$, with $\gamma_i = 1$ if $i = j$, 0 otherwise in Equation (1). And Equation (4) becomes

$$J_{\text{MM}}^{\text{orth}} = \sum_{v=1}^{V} \sum_{i=1}^{k} \left[ (\delta_i^{[v]} \gamma_i - (w^{[v]})^\top x_i^{[v]})\alpha_i - \frac{\lambda}{2}\alpha_i^2 \right] + \frac{1}{2} \sum_{v=1}^{V} \|w^{[v]}\|_{\text{F}}^2.$$

The orthogonal score derived by this new objective is

$$S_j^{[v]} = \delta_j \left[ K^{[v]}(K_{\text{MM}} + \lambda I)^{-1} \right]_{jj}.$$

We report the downstream performance of domain weights computed through Orthogonal score in Table 16. More importantly, Orthogonal score and Alignment score (MMix) are presented separately below. The performance of Alignment score used in MMix shows higher average accuracy than Orthogonal score. Importantly, Orthogonal score performs worse than UNIFORM on average (37.62 vs 38.00).

Table 15: **Transfer domain weights from LLaVA-0.5B to Qwen2-VL-2B for video-image-text instruction tuning.**

| Benchmark | UNIFORM | AVG | FUSED | **MMix** |
|---|---|---|---|---|
| AI2D | 67.78 | 67.94 | 67.29 | 68.26 |
| DocVQA | 75.10 | 75.76 | 80.11 | 78.11 |
| InfoVQA | 42.69 | 42.42 | 42.72 | 44.02 |
| MathVerse | 21.19 | 18.78 | 23.73 | 23.98 |
| MMBench | 59.02 | 56.44 | 55.33 | 59.71 |
| MMStar | 41.37 | 42.90 | 40.89 | 41.11 |
| MMMU | 37.44 | 36.00 | 35.98 | 37.44 |
| ScienceQA | 77.89 | 79.13 | 77.84 | 79.23 |
| OCRBench | 71.60 | 73.80 | 72.90 | 72.30 |
| RealworldQA | 58.82 | 58.82 | 58.04 | 58.69 |
| Video_MMMU | 21.02 | 21.50 | 20.32 | 20.83 |
| MVBench | 56.38 | 56.88 | 56.88 | 56.92 |
| Average | 52.53 | 52.53 | 52.67 | **53.38** |
| Number over UNIFORM | - | 7/12 | 6/12 | 8/12 |

Table 16: **Comparison of Orthogonal and Alignment scores for LLaVA-0.5B image-text instruction tuning.** Orthogonal score is even worse than UNIFORM (37.62 vs 38.00).

| Benchmark | Orthogonal score | Alignment score (**MMix**) |
|---|---|---|
| AI2D | 43.04 | 43.52 |
| DocVQA | 41.58 | 42.92 |
| InfoVQA | 21.49 | 22.13 |
| MathVerse | 15.99 | 18.91 |
| MMBench | 32.65 | 42.44 |
| MMStar | 34.52 | 35.88 |
| MMMU | 30.22 | 29.78 |
| ScienceQA | 64.25 | 64.50 |
| OCRBench | 46.30 | 45.80 |
| RealworldQA | 46.14 | 46.54 |
| Average | 37.62 | **39.24** |

## C FURTHER COMPARISONS WITH RELATED WORKS

**Data mixing in LMs.** Finding a high-quality data composition for LM pretraining is crucial for improved performance. Domain reweighting improves LM downstream performance by rebalancing data contributions from different sources (Brown et al., 2020; Touvron et al., 2023; Blakeney et al., 2024), but manual data mixing is not scalable and may lead to suboptimal domain weights (Albalak et al., 2024; Jiang et al., 2024; Aryabumi et al., 2024). Therefore, some works in the LM field explore the data mixing problems. DoReMi (Xie et al., 2023) employs a small proxy model to redistribute weights across various domains using Group DRO (Sagawa et al., 2020), thereby enhancing the training effectiveness of large base models. Group DRO was also used in (Thudi & Maddison, 2025). DoGE (Fan et al., 2024b;a) employs approximate bilevel optimization to train proxy models for domain weight determination. Recently, (Liu et al., 2024c) employs linear regression models to approximate validation loss across diverse data mixtures by training a large number of very small proxy models. Chen et al. (2024d) create a more general framework with the above methods as specific instantiations. Nevertheless, proxy-based methods necessitate algorithmic modifications in the training procedure, incurring supplementary proxy computational expenditure when multiple training stages are required, as is the case in VLMs. Moreover, these approaches are limited to small proxy models, which may not be feasible within the context of VLMs with both vision and language models. Other approaches focus on optimizing certain skills, e.g., Chen et al. (2023) introduced a skills-oriented framework for modulation of data mixtures during model training. Thudi et al. (2025)

use proxy models in a bilevel optimization framework to optimize the data mixture with downstream data samples. Held et al. (2025) propose mixing by estimating influence on downstream performance from each domain and assuming a linear model for the mixture weights. Another line of works featurize the datasets by deriving a compact domain representation, e.g., through clustering Zhang et al. (2025) or pooling Xie et al. (2025). The domain featureizations are then used to optimize dataset compositions, i.e., deciding weights to assign to the components of a combined dataset, through, e.g., correlation with validation set performance (Zhang et al., 2025) or through leverage scores (Xie et al., 2025). Drawing inspiration from scaling law research (Kaplan et al., 2020; Hoffmann et al., 2022), Data Mixing Laws (Ye et al., 2024) characterize the relationship between mixtures through exponential formulations, with other data mixture scaling laws proposed in (Que et al., 2024; Gu et al., 2024; Jiang et al., 2024; Kang et al., 2024). Overall, these works have shown that choosing the right data mixture in LMs can boost performance significantly in terms of perplexity and downstream tasks' accuracy. However, these works are limited to LMs and do not consider the challenges posed by VLMs, which require a more complex data mixture strategy due to, e.g., the multimodal nature of the data, missing modalities, and different training pipelines.

**Leverage score mixing.** Our unimodal scores measure the alignment of each domain w.r.t. the weight vector $w$ optimally aligned with the entire distribution, formulated through an alignment maximization task (1). Other common related learning tasks include ridge leverage scores (RLS). RLS measures the uniqueness of each data point through a weighted norm of the rows of the eigenvector matrix of the covariance. Specifically, RLS aims to find a vector $w$ being orthogonal to all data points except $x_i$. It can be formulated as regression for each domain $i$ separately with error variables $e_j = \gamma_j - w^\top x_j$, with $\gamma_j = 1$ if $j = i$, $0$ otherwise . Xie et al. (2025) assign higher weights to domains with lower RLS, thus employing inverse RLS as a proxy for dominant directions. In our scores, we formulate (1) that seeks $w$ exhibiting alignment with the *entire collection* of domain embeddings. This is achieved by assigning a uniform target value of 1 for all domains, i.e., $e_i = 1 - w^\top x_i$. Our scores thus have a different objective, which can be analyzed in the following perspectives. *(i)* Our resulting alignment score directly quantifies domain relevance, allowing for direct reweighting without inversion. This foundational difference in objective allows for a more natural and direct measure of alignment to the data domains. *(ii)* Our work aims to capture multi-modal couplings in the data domains. Our new direct formulation (1) facilitates the construction of the multi-modal objective. By introducing shared latent variables $\alpha_i$ via the Fenchel-Young inequality (2), we achieve principled coupling across multiple modalities in the dual formulation, whereas Xie et al. (2025) cannot easily achieve such extension through the inverse RLS.

**Data strategies for VLMs.** Data mixtures in VLMs are typically hand-picked by the model developers based on intuition or large grid searches, and no systematic approach is used to select the training data mixture. Qwen-VL (Bai et al., 2023b) employs a three-stage training pipeline utilizing a multilingual and multimodal corpus. The pre-training data is task-specific, e.g., captioning and OCR data. In the instruction tuning stage, they combine multi-modal and text-only dialogue to mantain language capabilities performance. LLaVA (Liu et al., 2023c; Li et al., 2024a; Liu et al., 2024a) additionally integrates LLM-generated instruction-following data with visual inputs. They openly release the LLaVA-OneVision (Li et al., 2024a) datasets as collections of domain-specific data, which we use in our experiments. Bunny (He et al., 2024) emphasizes the importance of high-quality data curation. Their approach focuses on finding coresets of the training dataset to improve model performance by removing uninformative image-text pairs. SAIL-VL (Dong et al., 2025) constructs a high-quality dataset through recaptioning via existing frontier VLMs. This curated dataset facilitates effective pretraining and fine-tuning of VLMs across various scales. Previous data selection works on CLIP training include, e.g., CiT (Xu et al., 2023), which proposes a dynamic data curation method coupling a data objective into the learning process by measuring the similarity between text embeddings and task-specific metadata; and, SIEVE (Mahmoud et al., 2024), which introduces a dataset pruning technique using synthetic captions generated by image-captioning models, allowing to identify and remove noisy or misaligned samples, enhancing dataset quality. Data strategies for VLMs have also been studied, e.g., data cleaning, toxicity removal, deduplication; see (Bai et al., 2024) for a comprehensive survey. DataComp (Gadre et al., 2023) deals with data filtering. Infinity-MM (Gu et al., 2025) investigates the scaling of multimodal models by increasing both model capacity and training data volume. W.r.t. integrating multiple modalities more in general, this is a long-standing challenge in machine learning (Baltrušaitis et al., 2019; Huang et al., 2023; Li et al., 2024d). Simple fusion methods, such as early fusion via concatenation (Barnum et al., 2020) or late

fusion by ensembling (Boulahia et al., 2021; Li & Tang, 2024), are often used. Another strategy is to learn a shared latent space where modalities are mapped to, enabling tasks like cross-modal retrieval (Liu et al., 2023c; Zhu et al., 2023), using contrastive learning (Radford et al., 2021; Alayrac et al., 2022) or duality (Houthuys et al., 2018; Tao et al., 2024). Other methods utilize attention to represent interaction between modality-specific encoders (Lu et al., 2019; Cai et al., 2024). Overall, the composition of training data is crucial for the performance of VLMs. To avoid reliance on expensive iterative performance measurements, our work introduces a method that can automatically assign appropriate resampling weights to each multi-modal domain of VLM training data.

## D    THE USE OF LARGE LANGUAGE MODELS (LLMS)

We utilize LLMs to assist in the preparation of this manuscript. The use of these tools was strictly limited to improving grammar, refining phrasing, and ensuring overall readability. The scientific contributions, including all ideas, methodologies, and analyses, are entirely our own.

