# OpenReview forum: "Multi-modal Data Mixtures for Vision-Language Model Training"
_ICLR.cc/2026/Conference — Submitted to ICLR 2026_

### Official Review · Reviewer_5Vc9 · 2025-10-30

**Soundness:** 2
**Presentation:** 3
**Contribution:** 2
**Rating:** 4
**Confidence:** 2

**Summary:**

As described in Introduction, the central question in this paper is to find out how to determine the optimal proportions of various domains for training multi-modal models. To this question, the authors propose to solve some optimization problem for finding an aligning vector $w^v$ for each modality $v$, and use it to compute scores $S^v_i$ for each modality $v$ and domain $i$, and finally compute the domain weight $p_i$ by softmax of the scores $\sum_v S^v_i$. In experiments, they show that the proposed method (marginally) outperforms the baselines of uniform or manual weighting, in multiple settings with modalities like texts, images, and videos.

**Strengths:**

- The paper is overall well-structured and easy to follow.
- The experimental design covered three modals, beyond the image-text pair.
- The proposed method seems to be simple to implement, and computationally feasible under a limited number of domains.
- Experimental results show marginal improvements compared to the baseline methods.

**Weaknesses:**

- It is overall unclear why the proposed method should work well. Particularly, it is unclear 1) why the exponential distribution with scores is expected to provide appropriate weights for each domain, 2) why we should compute these scores by the inner product between the domain's centroid $x_i$ and the aligned vector $w$, and 3) why the optimization problem eq.3 is appropriate to find aligning vectors for computing such scores. Or, another question is: 4) how the optimality of the eq.3 or eq.4 leads to the "optimal proportions" between domains (beyond subjective explanations).
- It is unclear how much the "modality-aware" weighting contributes to the improvement in accuracy by the proposed method, which is claimed as a core contribution of this work.
- The accuracy gain seems marginal even with additional weighting compared to the naive baselines. I think the proposed method may have almost no significance in practice, rather than collecting additional data or scaling up models.

**Questions:**

See weaknesses.

---

> ### Author Response · Authors · 2025-11-22
> **Response to Reviewer 5Vc9 (Part 1)**
>
> We thank the reviewer for the constructive comments. We address your concerns point-wisely below.
>
> > Q1. It is unclear why the proposed method should work well.
>
> Our work formulates the data mixing problem through the lens of modality-aware alignment maximization, which allows to efficiently compute closed-form scores directly via the Fenchel dual through the multi-modal kernel $K_{MM}$.
> The scores are converted into probability distributions $p$ by applying softmax normalization.
>
> While determining proportions that are optimal with respect to arbitrary downstream generalization losses remains an open challenge, our score can be interpreted from the spectral perspective.
> In fact, we can assert the following:
> Let $K_{\text{MM}} \in \mathbb{R}^{k \times k}$ be the multi-modal kernel matrix with SVD $K_{\text{MM}} = {U} {\Sigma} {U}^\top$, where $U=[u_1, \ldots, u_k]$ are the singular vectors and $\sigma_1 \geq \dots \geq \sigma_k$ are the singular values.
> The alignment score $S_i$ derived from the MMix objective (Proposition 3.1) is given by:
> $S_i = \sum_{j=1}^k \left( \frac{\sigma_j}{\sigma_j + \lambda} \right) ({u}_j^\top \delta) ({u}_j)_i$.
>
> Therefore, our score applies a spectral soft thresholding filter to the domain distribution.
> The alignment operator dampens the noisy directions (small eigenvalues) and measures the projection of domain $i$ onto the robust semantic subspace (corresponding to large eigenvalues).
>
> This relationship provides additional theoretical justification for our approach.
> We have added Appendix A.4 to explicitly link the data mixing goal to the spectral perspective and emphasize our contributions w.r.t. multi-modal alignment scoring rather than optimality in general.
>
> We have additionally conducted the following ablation to investigate the benefit of scores upweighting domains that align with the robust direction.
> We formulate an "Orthogonal Score" objective, which prioritizes the uniqueness of different domains i.e., downweighting domains with high alignment.
> Specifically for the domain $j$, set $e_i = \gamma_i-w^\top x_i$, with $\gamma_i=1$ if $i=j$, $0$ otherwise in Eq.(1).
> And Eq.(4) is revised to
> $$J_{\text{MM}}^{\text{orth}} = \sum_{v=1}^V \sum_{i=1}^k \Big[ (\delta_i^{[v]} \gamma_i - (w^{[v]})^\top x_i^{[v]}) \alpha_i - \frac{\lambda}{2} \alpha_i^2 \Big] + \frac{1}{2} \sum_{v=1}^V \|w^{[v]}\|_\mathrm{F}^2.$$
>
> The orthogonal score derived by this new objective is $$S_j^{[v]} = \delta_j \left[K^{[v]} (K_{MM}+\lambda I)^{-1}\right]_{jj}.$$
>
> We compare Orthogonal score against MMix (Alignment score) and the UNIFORM baseline with the same setup of Table 1.
> The Orthogonal score strategy performs even _worse_ than UNIFORM (37.62 vs 38.00), whereas MMix improves upon it. This demonstrates that prioritizing alignment is the preferable objective.
> Note that we have added this ablation study in Appendix B.10.
>
> |Benchmark|Orthogonal score |Alignment score (MMix)|
> |-|:-:|:-:|
> |AI2D|43.04|43.52|
> |DocVQA |41.58|42.92|
> |InfoVQA |21.49|22.13|
> |MathVerse|15.99|18.91|
> |MMBench |32.65|42.44|
> |MMStar|34.52|35.88|
> |MMMU |30.22|29.78|
> |ScienceQA|64.25|64.50|
> |OCRBench |46.30|45.80|
> |RealworldQA|46.14|46.54|
> |**Average**|37.62|**39.24**|
>
>
> > Q2. It is unclear how much the "modality-aware" weighting contributes to the improvement in accuracy by the proposed method.
>
> We emphasize that our approach realizes both late and early fusion: MMix learns modality-specific directions $w^{[v]}$, as well as shared latent variables $\alpha$ across modalities.
> To quantify the specific contribution of our "modality-aware" formulation, we include AVG and FUSED baselines.
> Specifically, AVG is an independent unimodal mixture averaged post hoc, which is a late-fusion baseline.
> FUSED represents an early-fusion scores from concatenating inputs, following standard data-mixing practice in non-multimodal LLM pretraining.
> Empirically, MMix consistently outperforms both AVG and FUSED on average performance. Moreover, FUSED usually shows similar performance to UNIFORM.
> The advantage of MMix over AVG becomes substantially larger in the complex tri-modal (video-image-text) setting.
> This demonstrates that modeling the interaction between modalities is essential for improving accuracy as the number of modalities increases.

---

> ### Author Response · Authors · 2025-11-22
> **Response to Reviewer 5Vc9 (Part 2)**
>
> > Q3. The accuracy gain seems marginal even with additional weighting compared to the naive baselines.
>
> We clarify that the practical significance of MMix lies in its ability to consistently improve the efficiency of VLM training with _negligible additional computational cost_.
> Our current baselines include a costly, non-scalable manual grid search (the "HUMAN" baseline). MMix matches and even surpasses this expert-tuned domain weights via a single, negligible computation (0.59 GPU hours).
> Beyond final accuracy, MMix shows speedup over both HUMAN and UNIFORM in Figure 2, with just 56% and 78% steps, respectively.
>
> The performance gap widens as task complexity increases.
> In the more demanding tri-modal image–text–video setting, MMix yields even larger gains over baselines, demonstrating its value scales with the difficulty of the multi-modal integration.
>
> Furthermore, we add an additional experiment testing MMix's domain weights on Qwen2-VL [1]. MMix shows improvement on this new setting as well, indicating its transferability across model types. The detailed results are in Q1 of Reviewer W7md or Appendix B.9 in our updated paper.
>
> [1] Wang, Peng, et al. "Qwen2-vl: Enhancing vision-language model's perception of the world at any resolution." arXiv preprint arXiv:2409.12191 (2024).

---

### Official Review · Reviewer_HbVQ · 2025-11-01

**Soundness:** 2
**Presentation:** 3
**Contribution:** 3
**Rating:** 4
**Confidence:** 3

**Summary:**

This paper addresses the multiple domain dataset mixing problem used to train vision-language models (VLMs), which integrate images and language by aligning image features with language tokens. While research exists on determining the domain mixing ratio for text-modal LLMs, VLMs have continued to rely on heuristics to determine resampling weights. The paper formulates this ratio determination as a problem of maximizing alignment with general structures across modalities and domains in latent space. This score is computed by solving a linear equation with regularization over the feature space of VLM hidden states. Furthermore, the paper extends this formulation to handle data with missing modalities. Experiments demonstrate that the proposed method efficiently improves performance compared to uniform weights or weights based on human heuristics.

**Strengths:**

- **S1.** This paper likely formulates the resampling weight determination problem for datasets in VLMs for the first time and proposes an algorithm to solve it.
- **S2.** The algorithm proposed in this paper can simultaneously handle both multi-modal and uni-modal data, featuring a simple structure that solves closed-form solutions using regularized linear equations.
- **S3.** Experiments conducted on relatively lightweight models such as 0.5B and 7B suggest that the proposed method reliably improves the weights by uniform sampling and human heuristic.

**Weaknesses:**

- **W1.** The relationship between alignment of general structures in latent space and generalization performance has not been theoretically explained. This suggests that maximizing alignment of general structures, upon which the proposed method relies, may not lead to improved generalization performance of VLMs.
- **W2.** The paper refers to the proposed method as an “automatic data mixing strategy,” but in reality, it requires human-defined domains, and there is insufficient discussion on how these domain definitions should be specified. For example, datasets like TextVQA [a] mix multiple domains such as OCR, document reading comprehension, and mathematical reasoning, and it remains unclear how to handle such composite capabilities.
- **W3.** The proposed method relies on the latent state representations of existing VLMs. This means it determines the dataset mixing ratio by measuring alignment at the pretraining stage. In other words, the alignment state may fluctuate across iterations or epochs, and weights computed at pretraining may become suboptimal at the midpoint of training. Furthermore, if the performance of VLMs at the pre-training stage is insufficient, they may fail to recognize domains and thus be unable to perform effective feature extraction.
- **W4.** The paper has not verified scalability for models larger than 7B. However, since this verification requires substantial computational resources, it is not essential for verifying the research question. On the other hand, it is highly important for practical applications to investigate generalization across models. To further strengthen this claim within the accessible computational environment of the paper, ideas include computing weights using a 7B model and comparing the differences with the 0.5B model's weights. Additionally, it might be beneficial to confirm the generalizability of VLMs by using others besides LLaVA, such as Qwen2.5-VL [b] or InternVL3.5 [c].
- **W5.** The proposed algorithm requires $O(k^3)$ computational complexity for a domain size $k$, which could become a bottleneck if we anticipate further increases in the number of composite domains, as discussed in W2.

[a] Singh, Amanpreet, et al. "Towards vqa models that can read." CVPR 2019.

[b] Bai, Shuai, et al. "Qwen2. 5-vl technical report." arXiv preprint arXiv:2502.13923 (2025).

[c] Wang, Weiyun, et al. "Internvl3. 5: Advancing open-source multimodal models in versatility, reasoning, and efficiency." arXiv preprint arXiv:2508.18265 (2025).

**Questions:**

Please response the concerns raised in the weaknesses section.

---

> ### Author Response · Authors · 2025-11-22
> **Response to Reviewer HbVQ (Part 1)**
>
> We thank the reviewer for the constructive comments and the appreciation on the practical relevance of our work. We address your concerns point-wisely below.
>
> > Q1. The relationship between the alignment of general structures in latent space and generalization performance has not been theoretically explained.
>
> Formulating data mixture as modality-aware alignment maximization enables efficient computation of closed-form resampling weights directly via the dual form through the multi-modal kernel $K_{MM}$.
>
> While alignment is not itself a generalization bound, it functions as a robust steering direction suppressing domain-specific noise.
> In fact, we can assert the following:
> Let $K_{\text{MM}} \in \mathbb{R}^{k \times k}$ be the multi-modal kernel matrix with SVD $K_{\text{MM}} = {U} {\Sigma} {U}^\top$, where $U=[u_1, \ldots, u_k]$ are the singular vectors and $\sigma_1 \geq \dots \geq \sigma_k$ are the singular values.
> The alignment score $S_i$ derived from the MMix objective (Proposition 3.1) is given by:
> $S_i = \sum_{j=1}^k \left( \frac{\sigma_j}{\sigma_j + \lambda} \right) ({u}_j^\top \delta) ({u}_j)_i$.
>
> Therefore, our score applies a spectral soft thresholding filter to the domain distribution.
> The alignment operator dampens the noisy directions (small eigenvalues) and effectively measures the projection of domain $i$ onto the robust semantic subspace (corresponding to large eigenvalues).
> Note that we have added this discussion in Appendix A.4.
>
> While we agree that establishing a formal VLM generalization bound remains a challenging open problem, extensive empirical results validate that these scores can improve VLM performance, even compared to expert-tuned weights.
>
> To further validate the relationship between the alignment score and generalization, we design **a new ablation study using an "Orthogonal Score"**, which prioritizes the uniqueness of different domains i.e., downweighting domains with high alignment.
> Specifically for the domain $j$, set $e_i = \gamma_i-w^\top x_i$, with $\gamma_i=1$ if $i=j$, $0$ otherwise in Eq.(1).
> And Eq.(4) becomes
> $$J_{\text{MM}}^{\text{orth}} = \sum_{v=1}^V \sum_{i=1}^k \Big[ (\delta_i^{[v]} \gamma_i - (w^{[v]})^\top x_i^{[v]}) \alpha_i - \frac{\lambda}{2} \alpha_i^2 \Big] + \frac{1}{2} \sum_{v=1}^V \|w^{[v]}\|_\mathrm{F}^2.$$
>
> The orthogonal score derived by this new objective is $$S_j^{[v]} = \delta_j \left[K^{[v]} (K_{\text{MM}}+\lambda I)^{-1}\right]_{jj}.$$
>
> We compare Orthogonal score against MMix (Alignment score) and the UNIFORM baseline with the same setup of Table 1.
> The Orthogonal score strategy performs even _worse_ than UNIFORM (37.62 vs 38.00), whereas MMix improves upon it. This demonstrates that prioritizing alignment is the preferable objective.
> Note that we have added this ablation study in Appendix B.10.
>
> |Benchmark|Orthogonal score |Alignment score (MMix)|
> |-|:-:|:-:|
> |AI2D|43.04|43.52|
> |DocVQA |41.58|42.92|
> |InfoVQA |21.49|22.13|
> |MathVerse|15.99|18.91|
> |MMBench |32.65|42.44|
> |MMStar|34.52|35.88|
> |MMMU |30.22|29.78|
> |ScienceQA|64.25|64.50|
> |OCRBench |46.30|45.80|
> |RealworldQA|46.14|46.54|
> |**Average**|37.62|**39.24**|
>
>
> > Q2. The paper refers to the proposed method as an “automatic data mixing strategy”, but it requires human-defined domains. There is insufficient discussion on how these domain definitions should be specified.
>
> We thank the reviewer for the feedback.
> As we discussed in the first paragraph in Section 2 "Related Works", data mixing is one specific step out of multiple stages in VLM data pipelines, and our work focuses on weighting the given pre-curated, skill-oriented domains.
> We would like to highlight that curating data into skill-oriented domains is a common step in data preparation to ensure a balanced set of capabilities.
> State-of-the-art VLMs rely on pre-curated, capability-oriented data groups, and this curation is an upstream design choice prior to the data-mixing problem.
> Our contribution focuses on the data mixing problem given the domain structure, and our results show that MMix yields superior mixtures even when the underlying domains are composite.
>
> Exploring automatic domain discovery, e.g., via clustering for datasets without domain labels, is an interesting direction for future work.
> In addition, we have revised the manuscript to more clearly state our contribution and avoid any potential confusion.

---

> ### Author Response · Authors · 2025-11-22
> **Response to Reviewer HbVQ (Part 2)**
>
> > Q3. Weights computed at pre-training may become suboptimal as alignment fluctuates, and pre-trained features may be insufficient.
>
> Regarding the sufficiency of feature extraction, the latent representations are from the mid-stage checkpoint of LLaVA-OneVision [Li et al., 2024a], which has already undergone extensive pre-training on massive image-text pairs and shows good performance as documented by the model authors.
>
> We additionally conducted a stability analysis by continuing training the checkpoint `lmms-lab/llava-onevision-qwen2-0.5b-mid-stage-a4` on its public mid-training data for 500 and 1000 additional steps and recomputed domain weights using our method.
> The results, presented in the table below, demonstrate that the domain weights remain stable throughout training.
> Note that we have added this ablation experiment in Appendix B.6.
>
> | Domain | Pretrained checkpoint | +500 steps | +1000 steps |
> |---|:-:|:-:|:-:|
> |General| 22.09| 21.65 | 22.59 |
> |Doc/Chart/Screen| 31.86|  30.12 | 29.96 |
> |Math/Reasoning| 16.63 | 18.74 | 18.82 |
> |General OCR| 15.66| 15.17 | 15.82 |
> |Language| 13.76 | 14.33 | 12.99 |
>
>
> > Q4. Additional experiments: 1) scalability for models larger than 7B, 2) generalization to other models, 3) comparing domain weights obtained by embeddings from 0.5B and 7B models.
>
> With respect to **scalability for model sizes**, we have conducted experiments on both 0.5B and 7B models. The larger model is computational infeasible for us.
>
> For **generalization to other models**, we test domain weights obtained from LLaVA-0.5B on Qwen-VL-2B [1] with the same setup of Section 4.2.
> We observe that the benefit of our MMix's weights also **applies to the different Qwen2-VL-2B architecture**.
> Note that we have added this experiment in Appendix B.9.
>
> | Benchmark               | UNIFORM | AVG | FUSED | MMix  |
> | - | :-: | :-: | :-: | :-: |
> | AI2D                    |  67.78 | 67.94 | 67.29 | 68.26 |
> | DocVQA                  |  75.10 | 75.76 | 80.11 | 78.11 |
> | InfoVQA                 |  42.69 | 42.42 | 42.72 | 44.02 |
> | MathVerse               |  21.19 | 18.78 | 23.73 | 23.98 |
> | MMBench                 |  59.02 | 56.44 | 55.33 | 59.71 |
> | MMStar                  |  41.37 | 42.90 | 40.89 | 41.11 |
> | MMMU                    |  37.44 | 36.00 | 35.98 | 37.44 |
> | ScienceQA               |  77.89 | 79.13 | 77.84 | 79.23 |
> | OCRBench                |  71.60 | 73.80 | 72.90 | 72.30 |
> | RealworldQA             |  58.82 | 58.82 | 58.04 | 58.69 |
> | Video_MMMU              |  21.02 | 21.50 | 20.32 | 20.83 |
> | MVBench                 |  56.38 | 56.88 | 56.88 | 56.92 |
> | **Average**             |  52.53 | 52.53 | 52.67 |**53.38**|
> | **Number over UNIFORM** |    -   | 7/12 | 6/12 |  8/12  |
>
> To check **the stability of domain weights across model scales**, we compare domain weights derived from LLaVA-0.5B and LLaVA-7B embeddings. The results, shown below, demonstrate that our weighting strategy remains consistent across different model sizes.
> We have added this ablation study in Appendix B.9.
>
> | Domain | LLaVA-0.5B | LLaVA-7B |
> |---|:-:|:-:|
> |General| 22.09| 21.15 |
> |Doc/Chart/Screen| 31.86|  29.91 |
> |Math/Reasoning| 16.63 | 19.64 |
> |General OCR| 15.66| 17.06 |
> |Language| 13.76 | 12.23 |
>
> > Q5. The proposed algorithm requires $O(k^3)$ computational complexity for a domain size $k$, which could become a bottleneck if we anticipate further increases in the number of composite domains.
>
> We claim that the computational cost of the alignment score is still negligible even with an increasing number of domains. We report the time of score computation with different numbers of domains in the table below. Even increasing to $10^4$ domains, it only requires 22s on a single A100 GPU.
> More importantly, the cost of manually tuned weights by human will increase dramatically.
>
> Note that we have added this empirical result in A.1.1.
>
> |Number of domains|Score computation (s)|
> |:-|:-:|
> |10|0.07|
> |100|0.09|
> |1000|0.48|
> |10000|21.85|
>
>
> [1] Wang, Peng, et al. "Qwen2-vl: Enhancing vision-language model's perception of the world at any resolution." arXiv preprint arXiv:2409.12191 (2024).

---

### Official Review · Reviewer_W7md · 2025-11-01

**Soundness:** 4
**Presentation:** 3
**Contribution:** 4
**Rating:** 6
**Confidence:** 4

**Summary:**

This paper presents a novel framework, named MMix, for automatically determining multimodal data mixtures for VLM training. The determination problem of data mixture is formulated as a modality-aware alignment maximization across domains. The multi-modal alignment scores obtained from the dual solution through inter-modal coupling variables. In the proposed framework, missing modalities can also be handled easily, allowing for the systematic integration of heterogeneous multi-modal data. In experiments, 0.5B and 7B VLMs are applied to various benchmarks. Then, it has been shown that MMix can achieve better accuracies with marginal computational cost.

**Strengths:**

The idea of deriving mixture weights from the dual solution of an alignment-maximization problem is a novel idea. And the detailed and careful derivation is provided in Appendix. It provides a theoretical pathway for extending data-mixing methods developed for LLMs to the multimodal domain.

This method proposes a way to handle a missing-modality problem naturally in the equation.

The computational cost is drastically reduced. This fact shows the practicability of the proposed method.

In 4.2, it has been demonstrated that MMix scales to more complex multi-modal settings such as VideoQA.

From the multiple aspects listed above, the proposed method is practical and useful as well as being supported by theoretical guarantees.

Ablation studies in B.6 also facilitate better understanding of the proposed method.

**Weaknesses:**

The experimental validation is done only with LLaVA-OneVision families. The authors may want to conduct more extensive experiments using other fundamental models to show their generality. Discussion on whether the weights can also generally be transferred to other fundamental models would also be helpful.

This paper can be reorganized and revised to facilitate better understanding of possible readers. For instance, the flowchart in Figure 1 and the subsections in section 3 do not align each other, which may confuse possible readers.

Some minor modification proposals (no need to reply)
- Table captions are sometimes confusing. For instance, that for Table 2 does not mention that the weights are transferred from 0.5B to larger 7B model (such description is given in the manuscript but not in the caption). Therefore, readers may be confused with weights dedicatedly designed for the larger model.
- The authors might want to conduct statistical significance test to claim the “best” performance. I am OK even if there is no significance.

**Questions:**

None

---

> ### Author Response · Authors · 2025-11-22
> **Response to Reviewer W7md**
>
> We thank the reviewer for the constructive comments and for recognizing the novelty and completeness of our methodology. We address your concerns point-wisely below.
>
> > Q1. The experimental validation is done only with LLaVA-OneVision families. That would be helpful to add 1) extensive experiments using other fundamental models and 2) a discussion on whether the weights can be transferred to other models.
>
> Thanks for the suggestions!
> We test domain weights obtained from LLaVA-0.5B on Qwen-VL-2B [1] with the same setup of Section 4.2.
> We observe that the benefit of our MMix's weights also **applies to the different Qwen2-VL-2B architecture**.
> Note that we have added this experiment in Appendix B.9.
>
>
> | Benchmark               | UNIFORM | AVG | FUSED | MMix  |
> | - | :-: | :-: | :-: | :-: |
> | AI2D                    |  67.78 | 67.94 | 67.29 | 68.26 |
> | DocVQA                  |  75.10 | 75.76 | 80.11 | 78.11 |
> | InfoVQA                 |  42.69 | 42.42 | 42.72 | 44.02 |
> | MathVerse               |  21.19 | 18.78 | 23.73 | 23.98 |
> | MMBench                 |  59.02 | 56.44 | 55.33 | 59.71 |
> | MMStar                  |  41.37 | 42.90 | 40.89 | 41.11 |
> | MMMU                    |  37.44 | 36.00 | 35.98 | 37.44 |
> | ScienceQA               |  77.89 | 79.13 | 77.84 | 79.23 |
> | OCRBench                |  71.60 | 73.80 | 72.90 | 72.30 |
> | RealworldQA             |  58.82 | 58.82 | 58.04 | 58.69 |
> | Video_MMMU              |  21.02 | 21.50 | 20.32 | 20.83 |
> | MVBench                 |  56.38 | 56.88 | 56.88 | 56.92 |
> | **Average**             |  52.53 | 52.53 | 52.67 |**53.38**|
> | **Number over UNIFORM** |    -   | 7/12 | 6/12 |  8/12  |
>
>
> > Q2. This paper can be reorganized and revised to facilitate a better understanding of possible readers.
>
> Thank the reviewer for the suggestion!
> We clarify that Figure 1 illustrates the practical execution pipeline, while Section 3 provides the theoretical derivation for our method.
> To prevent confusion, we have rewritten the first paragraph of Section 3 and we will further improve the readability of our paper.
>
> > Q3. Some minor modification proposals: 1) table captions and 2) statistical significance test.
>
> Thanks for the great advice! We have revised the caption of Table 2 in our paper, clarifying the transferred weight.
> We also rephrase details in Section 4. For example, avoiding "the best" phrasing or clarifying "the best" in terms of the _average_ performance.
> In addition, we clarify that we report the standard deviation in the tables and the standard deviation of Table 4 is shown in Appendix B.8.
>
> [1] Wang, Peng, et al. "Qwen2-vl: Enhancing vision-language model's perception of the world at any resolution." arXiv preprint arXiv:2409.12191 (2024).

---

### Official Review · Reviewer_SyKh · 2025-11-02

**Soundness:** 2
**Presentation:** 3
**Contribution:** 3
**Rating:** 4
**Confidence:** 3

**Summary:**

This paper proposes an automated multimodal data mixing method, MMix. Its core innovation lies in quantifying the importance of each domain in training through "modality-aware domain alignment scores", and dynamically adjusting the sampling weights accordingly. The theoretical basis of this method is established on the framework of maximizing domain alignment and coupling shared latent variables, formally deriving an unsupervised and scalable weight distribution mechanism.

**Strengths:**

The paper focuses on how to automatically allocate sampling weights for multi-domain and multi-modal data in VLM training without relying on manual parameter tuning or expensive grid search. It is clearly pointed out that there are problems such as modal deficiency, domain heterogeneity, and the non-scalability of manual parameter adjustment in VLM training. The problem settings are close to the actual needs of the industry.

**Weaknesses:**

- The paper assumes that all domains should be aligned in a shared and unified direction (i.e., the projection vector w), and weights are allocated by maximizing the alignment score. This assumption ignores the heterogeneity among domains. For instance, the semantic spaces of the Math/Reasoning and OCR domains may not be in the same direction at all. Forcing alignment could lead to excessive compression or misleading weighting. This assumption is similar to the idea of single-view PCA, but in multimodal tasks, different domains may correspond to multiple independent semantic subspaces rather than a globally shared direction.

- The kernel matrix is simply summed by elements $K_{MM} = \sum_v K^{[v]}$. This linear superposition method ignores the scale differences and semantic inconsistencies between modes. For instance, the embedding dimension or norm of an image modality may be much larger than that of a text, resulting in its dominant kernel matrix.

- The VLM models used in the experiment are all out-of-dated. LLaVA is a 2023 model. It is recommended to use newer LMs for the experiment, which will be more convincing.

**Questions:**

Please respond to the theoretical issues listed in Weaknesses.

---

> ### Author Response · Authors · 2025-11-22
> **Response to Reviewer SyKh (Part 1)**
>
> We thank the reviewer for the constructive comments and the appreciation on the practical relevance of our work. We address your concerns point-wisely below.
>
> > Q1. The paper assumes that all domains should be aligned in a shared and unified direction, which ignores the heterogeneity among domains.
>
> Our modelling in Eq.(1) seeks maximal alignment, which is the _objective_ of the optimization rather than an _assumption_ that domains are colinear.
> In fact, if one had assumed perfect a priori alignment, the resulting mixture would correspond to the UNIFORM baseline.
> The empirical underperformance of UNIFORM (Tables 1, 2, 4) validates that indeed a non-uniform weighting is necessary.
>
> We would like to clarify that **heterogeneity among domains is not ignored**.
> Instead, upweighting domains whose embeddings are better aligned can benefit other domains as well.
> For instance, although MMix downweights Math and OCR relative to UNIFORM, it nonetheless shows competitive performance on MathVerse and OCRBench, indicating positive transfer from high-alignment domains (see Table 1 and the discussion in Section 4.1 line 336).
>
> Our multi-modal formulation explicitly avoids a single shared direction.
> In the multi-modal case, we do not learn a single projection vector $w$. Instead, we learn a *set* of distinct projection vectors ${w^{[v]}}$, one for each modality $v$ (Eq.(3)).
> The objective $J_{MM}$ in Eq.(3) jointly optimizes the modality-specific alignments with the shared latent variables $\alpha$.
> Therefore, **the intra- and the inter-modality relations are both explored**.
>
> To further address the reviewer's concern, we design **a new ablation study using an "Orthogonal Score"**, which prioritizes the uniqueness of different domains, i.e., downweighting domains with high alignment.
> Specifically for the domain $j$, set $e_i = \gamma_i-w^\top x_i$, with $\gamma_i=1$ if $i=j$, $0$ otherwise in Eq.(1).
> And Eq.(4) becomes
> $$J_{\text{MM}}^{\text{orth}} = \sum_{v=1}^V \sum_{i=1}^k \Big[ (\delta_i^{[v]} \gamma_i - (w^{[v]})^\top x_i^{[v]}) \alpha_i - \frac{\lambda}{2} \alpha_i^2 \Big] + \frac{1}{2} \sum_{v=1}^V \|w^{[v]}\|_\mathrm{F}^2.$$
>
> The orthogonal score derived by this new objective is $$S_j^{[v]} = \delta_j \left[K^{[v]} (K_{\text{MM}}+\lambda I)^{-1}\right]_{jj}.$$
>
> We compare Orthogonal score against MMix (Alignment score) and the UNIFORM baseline with the same setup of Table 1.
> The Orthogonal score strategy performs even _worse_ than UNIFORM (37.62 vs 38.00), whereas MMix improves upon it. This demonstrates that prioritizing alignment is the preferable objective.
> Note that we have added this ablation study in Appendix B.10.
>
> |Benchmark|Orthogonal score |Alignment score (MMix)|
> |-|:-:|:-:|
> |AI2D|43.04|43.52|
> |DocVQA |41.58|42.92|
> |InfoVQA |21.49|22.13|
> |MathVerse|15.99|18.91|
> |MMBench |32.65|42.44|
> |MMStar|34.52|35.88|
> |MMMU |30.22|29.78|
> |ScienceQA|64.25|64.50|
> |OCRBench |46.30|45.80|
> |RealworldQA|46.14|46.54|
> |**Average**|37.62|**39.24**|
>
> > Q2. The kernel matrix is simply summed by elements $K_{MM} = \sum_v K^{[v]}$. This linear superposition method ignores the scale differences and semantic inconsistencies between modes.
>
> We appreciate the reviewer’s insightful comment.
> The additive construction is in fact a well-established mechanism in multi-modal extensions of kernel methods [1].
> In our dual objective (Eq.(4)), the multimodal kernel $K_{MM}$ arises as the per-modality Gram matrices aggregate by sharing the same dual variables.
> While future work may explore combinations such as tensor products to model multiplicative interactions, the additive couplings remain an efficient and theoretically robust method for integrating multiple modalities.
>
> Regarding the modality norms, our formulation and implementation mitigate this issue.
> First, domain embeddings are centered in feature space to remove global offsets.
> Second, the embeddings $x^{[v]}$ for both text and vision modalities are extracted from the same hidden layer of the transformer, so they have identical dimensionality.
> In our experiments, we observed that the norms of the image and text kernels were comparable. If the scales differed significantly, the problem can be normalized by considering $K^{[v]}/Tr(K^{[v]})$. This is a well-studied normalization technique that is widely used in multiple kernel learning [2], and corresponds to equalizing the spectral energy of each modality's kernel.
>
> [1] Xu, C., Tao, D., & Xu, C. (2013). A survey on multi-view learning. arXiv preprint arXiv:1304.5634.
>
> [2] Bach, F., Thibaux, R., & Jordan, M. (2004). Computing regularization paths for learning multiple kernels. Advances in neural information processing systems.

---

> ### Author Response · Authors · 2025-11-22
> **Response to Reviewer SyKh (Part 2)**
>
> > Q3. The VLM models used in the experiment are all out-of-dated. LLaVA is a 2023 model. Recommend using newer VLMs.
>
> We clarify that we use the updated LLaVA-OneVision [Li et al., 2024a] from 2024 in our experiments, not the original LLaVA from 2023.
>
> To further test MMix's domain weights on other model architectures, we conduct additional experiments with `Qwen/Qwen2-VL-2B` [3] with the same setup of Section 4.2.
> We observe that the benefit of our MMix's weights also **applies to the different Qwen2-VL-2B architecture**.
> Note that we have added this experiment in Appendix B.9.
>
> | Benchmark               | UNIFORM | AVG | FUSED | MMix  |
> | - | :-: | :-: | :-: | :-: |
> | AI2D                    |  67.78 | 67.94 | 67.29 | 68.26 |
> | DocVQA                  |  75.10 | 75.76 | 80.11 | 78.11 |
> | InfoVQA                 |  42.69 | 42.42 | 42.72 | 44.02 |
> | MathVerse               |  21.19 | 18.78 | 23.73 | 23.98 |
> | MMBench                 |  59.02 | 56.44 | 55.33 | 59.71 |
> | MMStar                  |  41.37 | 42.90 | 40.89 | 41.11 |
> | MMMU                    |  37.44 | 36.00 | 35.98 | 37.44 |
> | ScienceQA               |  77.89 | 79.13 | 77.84 | 79.23 |
> | OCRBench                |  71.60 | 73.80 | 72.90 | 72.30 |
> | RealworldQA             |  58.82 | 58.82 | 58.04 | 58.69 |
> | Video_MMMU              |  21.02 | 21.50 | 20.32 | 20.83 |
> | MVBench                 |  56.38 | 56.88 | 56.88 | 56.92 |
> | **Average**             |  52.53 | 52.53 | 52.67 |**53.38**|
> | **Number over UNIFORM** |    -   | 7/12 | 6/12 |  8/12  |
>
> [3] Wang, Peng, et al. "Qwen2-vl: Enhancing vision-language model's perception of the world at any resolution." arXiv preprint arXiv:2409.12191 (2024).

---

### Author Response · Authors · 2025-11-28
**Summary of rebuttal as deadline is approaching**

Dear Reviewers,

Thank you for your constructive feedback. We highlight the main points of the rebuttal below, with detailed responses in the individual replies:

- **Alignment justification**:
We derived a spectral analysis showing that MMix acts as a spectral soft-thresholding filter on the multi-modal kernel (Appendix A.4).
Empirically, we performed a new ablation with the “Orthogonal Score” prioritizing diversity (Appendix B.10).
The results show that MMix significantly outperforms orthogonal weighting, supporting alignment as the better objective.

- **Evaluation on different models**:
We have extended our evaluation to Qwen2-VL-2B.
The results (Appendix B.9) show that MMix weights improve performance across different model architectures.
We clarified that our LLaVA experiments use the more recent 2024 version.

- **Practical significance**:
MMix replaces expensive grid-based tuning with a single cheap computation (0.59 GPU hours), and meanwhile it trains 1.28× faster in image-text tasks.
The performance gap widens in the complex video-image-text setting, where manual tuning becomes prohibitively expensive.

- **Computational complexity**:
We provided empirical timings showing that score computation is negligible (taking just 22s on a single A100 even with $10^4$ domains) and that the whole reweighting pipeline costs a tiny fraction of the underlying VLM training budget.

- **Expanded ablation studies**:
To further validate the stability of our method, we have extended our ablation studies (Appendix B.6). New experiments demonstrate that MMix's domain weights remain stable across model sizes and training steps.

- **Scope and presentation improvements**:
We have reorganized Section 3 to improve readability, refined table captions, and clarified that our method optimizes weights for a given set of capability-oriented domains, consistent with standard VLM pipelines.

We believe the provided clarifications and additional experiments strengthen the paper's contributions. We hope this clarifies the merits of our work. As the discussion window is closing soon, we would be happy to address any further questions you may have.

Best regards,

The Authors

---

### Author Response · Authors · 2025-12-01
**Summary of rebuttal for AC**

Dear Program Chairs, Area Chairs, and Reviewers,

Given the exceptional early cancellation of the discussion period, we summarize below the strengths of the paper and our responses to the main concerns from the reviewers, which we hope will be helpful for the final decision.

**Strengths & reviewers’ support:**
Reviewers all recognized the *novelty* of our framework formulating VLM data mixing as a multi-modal alignment maximization problem, and its significant practical value in *realistic industrial scenarios*.
- Reviewer W7md with the highest confidence (4), supports our paper (Score: 6), considering the "theoretical pathway" and "drastically reduced computational cost" as excellent contributions.
- Reviewer HbVQ acknowledges that this work "likely formulates the resampling weight determination problem... for the first time" and praises the algorithm's ability to "simultaneously handle both multi-modal and uni-modal data".
- Reviewer SyKh notes the problem settings are "close to the actual needs of the industry" and effectively address the non-scalability of manual parameter tuning.
- Reviewer 5Vc9 agrees the method is "computationally feasible" and "simple to implement".

**Main concerns from the reviews:**
In the rebuttal, we have responded to each comment and addressed the concerns with new experiments and theoretical analysis.
Below, we summarize the main points.

1. *Generalization across architectures* (SyKh, W7md, HbVQ):
We extended our evaluation to **Qwen2-VL-2B (Appendix B.9)**, showing our derived weights transfer successfully and yield performance gains across different model families.
2. *Validation of the alignment objective* (SyKh, 5Vc9, HbVQ):
We provided a deeper analysis of the methodology.
We added (1) a  **spectral analysis (Appendix A.4)**, showing that MMix acts as a spectral soft-thresholding filter, and (2) an **orthogonal score ablation (Appendix B.10)**, where we tested the alternative scoring method prioritizing uniqueness. MMix significantly outperformed it, supporting that our alignment objective is the better approach.
3. *Practical significance* (5Vc9):
The value of MMix is not just for the accuracy gains but also for **efficiency** w.r.t.: (1) **Scalability**: Current practices rely on manual grid searches, which are computationally expensive and unscalable as the number of modalities increases. MMix replaces this with a negligible one-time computation (**0.59 GPU hours**, Table 3); and (2) **Training efficiency**: MMix achieves expert-tuned performance **1.28x faster** (Figure 2, Right).

We believe these additions further strengthen MMix as a practically useful solution for real-world and industrial VLM training, and we sincerely thank the reviewers and chairs for their time and constructive comments.

Best regards,

The Authors

---

### Meta-Review · Area_Chair_JrVq · 2025-12-12

**Summary:**

The paper aims at formulating a multimodal data mixing method that estimates the importance of each domain in training through "modality-aware domain alignment scores", and dynamically adjusting the sampling weights accordingly.

Strengths identified by reviewers include the paper's writing and its novel in realistic industrial scenarios.

However, there are two main concerns shared across three reviewers (SyKh, HbVQ, 5Vc9) and have not been fully addressed: 1) aligning all domains into a globally shared space would ignore the heterogeneity among domains; and 2) relations of aligning all domains into a globally shared space and generalization performance is not clearly explained.

Given these considerations, this paper clearly fails to meet the standards of ICLR. The authors are encouraged to address the highlighted issues to strengthen their contribution to the field of VLM.

**Reviewer Concerns:**

The authors provided detailed feedback and most of concerns have been addressed, including a) the summation of kernel matrix may ignore the scale and semantic difference; b) out-of-date compared methods; c) the method relies on predefined domain information and can not automatically identify domain information when the dataset consists of different forms of data such as OCR, document reading comprehension, and mathematical reasoning; d) weights computed from the pertaining stage may be suboptimal for the midpoint of training; e) generalization across architectures; f) potentially high computational cost of the method; g) limited accuracy gains of the proposed weighting methods compared to including additional data etc; h) typos.

However, two main concerns are still not fully addressed: 1) aligning all domains into a globally shared space would ignore the heterogeneity among domains; and 2) relations of aligning all domains into a globally shared space and generalization performance is not clearly explained.

**Reviewer Scores:**

The initial reviewer scores are mixed (three borderline reject and one borderline accept). As two main concerns are not fully addressed, reviewers would have maintained or decrease their scores after rebuttal.

---

### Decision · Program_Chairs · 2026-01-26

Reject